# Diet-omics in the S̲tudy of U̲rban and R̲ural C̲rohn disease E̲volution (SOURCE) cohort

Tzipi Braun[1,16], Rui Feng[2,3,16], Amnon Amir [1], Nina Levhar[1,4], Hila Shacham[1,4], Ren Mao[2], Rotem Hadar [1], Itamar Toren[1,5], Yadid Algavi [4], Kathleen Abu-Saad [6], Shuoyu Zhuo[2], Gilat Efroni[1], Alona Malik[1], Orit Picard[1], Miri Yavzori[1], Bella Agranovich[7], Ta-Chiang Liu[8], Thaddeus S. Stappenbeck [9], Lee Denson [10], Ofra Kalter-Leibovici [4,6], Eyal Gottlieb [11], Elhanan Borenstein [4,12,13], Eran Elinav [14,15], Minhu Chen[2,16], Shomron Ben-Horin[1,4,16] & Yael Haberman [1,4,10,16] ✉

Crohn disease (CD) burden has increased with globalization/urbanization, and the rapid rise is attributed to environmental changes rather than genetic drift. The Study Of Urban and Rural CD Evolution (SOURCE, n = 380) has considered diet-omics domains simultaneously to detect complex interactions and identify potential beneficial and pathogenic factors linked with rural-urban transition and CD. We characterize exposures, diet, ileal transcriptomics, metabolomics, and microbiome in newly diagnosed CD patients and controls in rural and urban China and Israel. We show that time spent by rural residents in urban environments is linked with changes in gut microbial composition and metabolomics, which mirror those seen in CD. Ileal transcriptomics highlights personal metabolic and immune gene expression modules, that are directly linked to potential protective dietary exposures (coffee, manganese, vitamin D), fecal metabolites, and the microbiome. Bacteria-associated metabolites are primarily linked with host immune modules, whereas diet-linked metabolites are associated with host epithelial metabolic functions.

Crohn disease (CD) burden has increased with globalization[1]. This rapid increase is likely linked to environmental and dietary changes[2], leading to alterations in the human gut microbiome and the immune and epithelial intestinal mucosa, which in turn are thought to instigate chronic gut inflammation in CD. Despite significant medical advances, the majority of CD patients do not enjoy complete control of disease activity and symptoms using current therapies[3]. This partly stems from gaps in knowledge of the causes of this complex disease which preclude the development of targeted therapies.

[1]Sheba Medical Center, Tel-Hashomer, Affiliated with the Tel Aviv University, Tel Aviv, Israel. [2]The First Affiliated Hospital, Sun Yat-Sen University, Guangzhou, Guangdong, China. [3]Department of Gastroenterology, Guangxi Hospital Division of The First Affiliated Hospital, Sun Yat-Sen University, Nanning, Guangxi, China. [4]Faculty of Medical and Health Sciences, Tel Aviv University, Tel Aviv, Israel. [5]Department of Military Medicine, Faculty of Medicine, Hebrew University of Jerusalem, Jerusalem, Israel. [6]Gertner Institute for Epidemiology and Health Policy Research, Tel Hashomer, Israel. [7]Laura and Isaac Perlmutter Metabolomics Center, Technion-Israel Institute of Technology, Bat Galim, Haifa, Israel. [8]Department of Pathology and Immunology, Washington University in St Louis School of Medicine, St. Louis, MO, USA. [9]Department of Inflammation and Immunity, Lerner Research Institute, Cleveland Clinic, Cleveland, OH, USA. [10]Cincinnati Children's Hospital Medical Center, Department of Pediatrics, University of Cincinnati College of Medicine, Cincinnati, OH, USA. [11]The Ruth and Bruce Rappaport Faculty of Medicine, Technion-Israel Institute of Technology, Bat Galim, Haifa, Israel. [12]Blavatnik School of Computer Science, Tel Aviv University, Tel Aviv, Israel. [13]Santa Fe Institute, Santa Fe, NM, USA. [14]Department of Systems Immunology, Weizmann Institute of Science, Rehovot, Israel. [15]Microbiome & Cancer Division, German National Cancer Center (DKFZ), Heidelberg, Germany. [16]These authors contributed equally: Tzipi Braun, Rui Feng, Minhu Chen, Shomron Ben-Horin, Yael Haberman. ✉e-mail: yael.haberman@cchmc.org

Efforts to delineate some of the environmental and dietary factors driving CD in the West have been hampered by near-universal modernization. However, in China, the transition from a traditional, predominantly rural society to an urban and industrialized one is currently ongoing, paralleling a staggering rise in the incidence of CD, a disease that was barely known in China two decades ago[4]. This provides a unique opportunity for exploring different aspects of CD evolution in China[5] and comparing it to CD in the West. For example, a metanalysis that compared environmental and dietary risk factors between Eastern and Western populations highlighted that higher fat intake, higher monounsaturated fatty acid (MUFA), n-3 polyunsaturated fatty acid (PUFA), and n-6 PUFA are associated with CD only in the Eastern population[6].

CD human cohorts and biospecimens are key for understanding factors contributing to CD[7–10] as existing disease models do not fully recapitulate all human aspects of the disease. Therefore, to provide insights into the impacts of urbanization and modernization and to decipher CD pathogenesis in the gut, we structured the Study Of Urban and Rural Crohn disease Evolution (SOURCE), a multicenter and multiomics cross-sectional study in Guangdong province in China and in Israel assessing 380 newly diagnosed CD patients and controls living in rural and urban settings. Prospectively collected data included demographics, clinical characteristics, dietary exposures based on a food frequency questionnaire (FFQ), environmental exposures (childhood factors, dietary and smoking habits, and sanitary conditions), fecal microbiome, ileal transcriptome, and fecal metabolome. Here we show that this unique cohort design enabled us to reach biologically significant insights including (1) Rural-to-urban transition mirrors some changes also seen in CD, (2) Specific dietary exposures were linked with gut microbial taxa previously enriched in CD, (3) By applying an unbiased approach to the mucosal ileal transcriptomics data, we defined gene co-expression modules that were linked to potential beneficial dietary exposures, fecal metabolites, and the microbiome.

## Results

### Rural and urban populations differ in exposures

The SOURCE cohort study was conducted in Guangdong province in Southern China [Sun Yat-Sen (SYS) First Affiliated Hospital, China] and in Israel (Sheba Medical Center, Israel) and included 380 participants (Fig. 1a). In China; 40 newly-diagnosed CD patients and 121 healthy residents of Guangzhou (urban Ctl), a modernized metropolitan community with a population of 16 million, and 162 healthy residents of Shaoguan district (all rural controls), a rural underdeveloped community 300 km north of Guangzhou. In Israel; 25 newly-diagnosed pre-treatment CD patients and 32 healthy controls. Demographics, CD phenotype, biomarkers (CRP and fecal calprotectin), food frequency questionnaire (FFQ), the environment questionnaire developed by the International Organization of IBD (IOIBD), microbiome V4 16S sequencing (stool and mucosal biopsies), microbial metagenomics shotgun sequencing (MGX), ileal transcriptomics (biopsies), and stool metabolomics were included (Fig. 1b). Subjects from the Israeli cohort (n = 57) were all White and mostly lived in urban settings (95.5% CD, 88% Ctl). Subjects from China (n = 323) were all Asians. Total energy consumption (Kcal/day) did not differ between groups (Table 1). Age was not different between CD and Ctl in Israel and urban Chinese, but the overall rural Ctls (n = 162) were older, and this was taken into consideration in the analyses. As we wanted to evaluate urban exposures, we specifically added a question to the IOIBD that specified the amount of time spent in an urban environment in the last year (less than 10%, 10–50%, ≥50%). To refine the data analysis and based on the environmental questionnaire, we stratified all subjects living in a rural area (all rural, n = 162) to those spending less than 50% of their time in the last year in urban areas (designated "rural", n = 88) and compared them to those living in a rural area but spending ≥50% of their time in

an urban environment (designated "rural-urban", n = 74), and to city-dwellers ("urban", n = 121), and the newly diagnosed CD patients (n = 40). Environmental exposures differed between groups (Fig. 1c and Supplementary Dataset 1). For example, flush toilet availability was reported in 5% of the rural group, 35% of the rural-urban group, 61% of urban participants, and 100% of Israelis. Having farm animals was noted in 60% of rural, 24% of rural-urban, and 3% of urban participants. Dietary habits also differed substantially; drinking soft drinks at least weekly was reported in 5% of rural and 27% of urban Chinese subjects, and 60% of the Israeli controls, and similarly coffee was reported in 1% of rural, 15% of rural-urban, and 32% urban Chinese, and 77% of Israelis. Smoking differed by group and gender (higher rates in males), with negligible percentages of active female smoking in China.

### Time spent in urban environments affect microbiome and metabolites

Among rural Chinese participants, unsupervised Principal Coordinates Analysis (PCoA) analysis using the unweighted UniFrac metric indicated time spent in the city (i.e., rural vs. rural-urban) was a major factor driving differences in the healthy rural microbiome (Fig. 2a), with rural controls predominantly clustered on one side of the plot and rural-urban concentrated on the other side. Indeed, we observed differences in UniFrac based beta-diversity (Fig. 2b) and alpha-diversity (Fig. 2c), with significantly lower diversity in rural-urban individuals vs. rural controls. Further, a previously defined microbial health index[11] was numerically reduced in the rural-urban participants vs. rural controls (Fig. 2d). To validate whether the differences in microbial communities are indeed related to rural/urban exposure, we used a dataset from an independent cohort[5] (BioProject PRJNA349463) that included subjects from Hunan province in Southern China (Fig. 2e), and based on it we generated an independent "rural index" on bacterial amplicon sequence variants (ASVs) that showed significant differential abundance between rural and urban cases from that cohort[5]. When this "rural index" was applied to our cohort, it showed significantly lower read-out in rural-urban participants vs. rural controls, and was also lower in CD cases vs. urban controls (Fig. 2f). Similar differences in rural index and health index between rural-urban vs. rural controls were noted after age-matching these groups (Supplementary Fig. 1a–e). Finally, MaAsLin multivariate analyses (controlling for age and gender) identified 41 ASVs higher in rural-urban individuals (Fig. 2g, h shows those with p < 0.005 and FDR ≤ 0.1, and Supplementary Dataset 3 shows taxa with p ≤ 0.05, FDR ≤ 0.25) compared to rural controls, including several *Bacteroides*, *R. gnavus*, and *Fusobacteriaceae*, that were previously linked with CD[7]. In contrast, the rural community showed enrichment in 37 ASVs including *Actinomyces* and *Bifidobacterium*. dbBact-based enrichment analysis of the ASVs higher in rural participants indicated that such enrichments had previously been seen in other rural communities (Supplementary Fig. 1).

PCoA based on the fecal metabolites (Fig. 2i) indicated that time spent in the city (rural vs. rural-urban) was a major factor driving differences within the healthy rural participants, which did not differ in age. MaAsLin multivariate analyses identified 22 metabolites that differed between rural-urban and rural individuals, 8 that were increased and 14 that were decreased after controlling for age and gender (Fig. 2j and Supplementary Dataset 3 with p < 0.01, FDR ≤ 0.25). Overlapping these metabolites with those that were significantly different between CD and urban control Chinese (Fig. 2j, k) indicated that 8 of 8 (100%) metabolites higher in rural-urban individuals were also significantly increased in CD, including N-acetyltryptophan, N-acetylalanine, and oleic and palmitoleic acids. In contrast, 12 of the 14 (90%) reduced metabolites in rural-urban individuals vs. rural individuals were also significantly reduced in CD, including phenylpyruvate, glutarate, and aminoadipate. Spearman correlation on the coefficient of these two independent comparisons showed a significant correlation (r = 0.902, p = 9.6E−9, Fig. 2k). The microbial health and rural index and the high

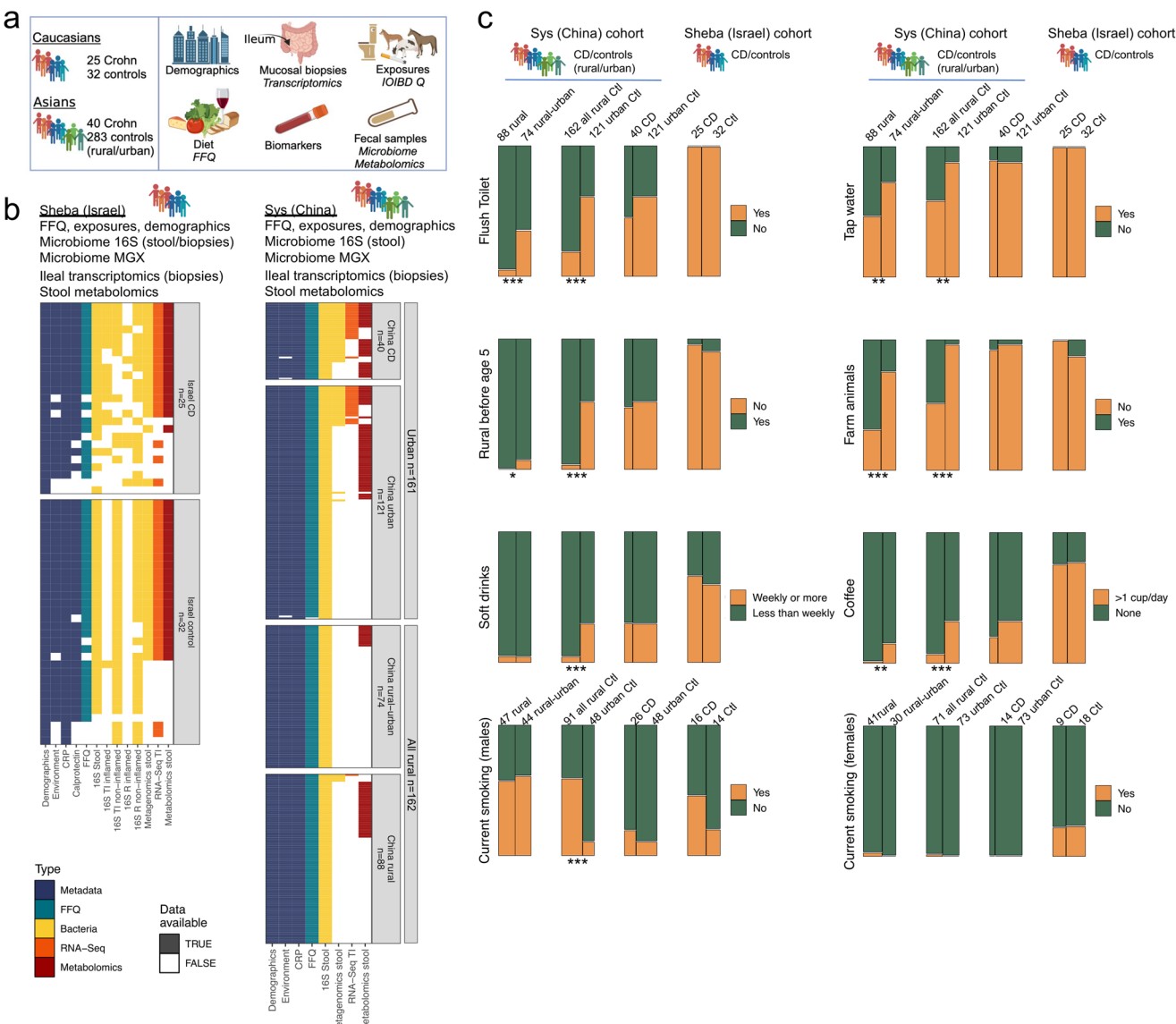

**Fig. 1 | Diet-omics SOURCE cohort demographics, sampling, and dietary and environmental exposures.** The SOURCE cohort included 380 participants. In China, 40 newly-diagnosed Crohn disease (CD) patients and 121 healthy residents of Guangzhou (urban Ctl) and 162 healthy residents of Shaoguan district (all rural controls), a rural underdeveloped community 300 km north of Guangzhou. In Israel, 25 newly-diagnosed pre-treatment CD patients and 32 healthy controls. **a** Scheme (BioRender) illustrating SOURCE demographics, sampling, and metadata including exposures. **b** Cohort figure showing Israel and China data types and availability colored as indicated. Each row represents a subject. **c** Mosaic plots showing exposure rates between the different groups in China, and Israel. Full data in Supplementary Dataset 1. Two-sided *$p < 0.05$, **$p < 0.01$, ***$p < 0.001$, chi-square tests test. IOIBD Q International Organization of IBD questionnaire, FFQ food frequency questionnaire, MGX metagenomics, TI terminal ileum, Ctl controls.

correlation coefficient results of the metabolomics dataset indicated that gut microbiome and metabolome observed in this rural-urban transitional group mirrored changes seen also in CD.

To capture the complex interactions in the gut among all rural subjects ($n = 162$), which showed higher heterogeneity in exposures including diet, we applied HAllA analysis[12]. This indicated significant associations between 16S and diet ($n = 162$, $p < 0.005$, FDR ≤ 0.25, Supplementary Fig. 1g and Supplementary Dataset 3); *R. gnavus* was positively linked with fat (monounsaturated fatty acids) and fish and poultry consumption, while *Oxalobacter formigenes* was negatively associated with these factors, and specific *Clostridium* taxa were linked with whole grains servings. We also detected 2675 significant associations between microbiome and the fecal metabolome [$n = 40$, $p < 0.005$, FDR ≤ 0.25, 684 significant associations with $p < 0.0005$, FDR < 0.1 (Supplementary Fig. 1g and Supplementary Dataset 3)]; here *R. gnavus* was positively associated with docosatetraenoic acid,

histidine, and docosapentaenoic acid (DPA), while azelate, which is currently used to suppress inflammation in the skin[13], was linked with *Oxalobacter formigenes*. We identified fewer or no significant correlations between metagenomic shotgun taxonomy, diet, and metabolites, likely due to the reduced number of available samples.

### Exposures and diet are linked with microbial variations

We next used PCA on the dietary exposures to highlight factors discriminating between samples and to explain the variation in dietary exposure in the overall cohort in Israel and China and supplemented this with PERMANOVA aiming to quantify the contribution of different factors affecting the gut microbial composition within each subgroup separately (rural, rural-urban, urban and CD in China, and controls and CD in Israel), after controlling for age and gender. PCA based on dietary variables was used to capture dietary patterns between subjects and groups in Israel and China based on data collected within

**Table 1 | SOURCE basic demographics and characteristics**

| | SYS sub-cohort (n = 323) 100% Asian ethnicity | | | Sheba sub-cohort (n = 57) 100% White ethnicity | |
|---|---|---|---|---|---|
| | Ctl all rural (n = 162) | Ctl urban (n = 121) | CD (n = 40) | Ctl (n = 32) | CD (n = 25) |
| Age (years) | 51 (43, 59)# | 27 (25, 31) | 27 (20, 30) | 28 (23, 45) | 30 (24, 41) |
| Female gender (%) | 71 (44%) | 73 (60%) | 14 (35%) | 18 (56%) | 9 (36%) |
| BMI | 22 (20, 24) | 20 (19, 21) | 17 (16, 20)^ | 24 (22, 27) | 22 (20, 27) |
| CRP (normal < 5) | 0.82 (0.4, 1.9) | 0.3 (0.2, 0.7) | 16 (4, 44)^ | 2 (1, 6) | 11 (4.6, 35)* |
| Calprotectin | – | – | – | 57 (35, 227) | 1000 (731, 1000)* |
| CD Location | | | n = 37 | | n = 24 |
| Ileal (L1) | – | – | 13 (35%) | – | 15 (63%) |
| Colom (L2) | – | – | 4 (11%) | – | 1 (4%) |
| Ileal + colon (L3) | – | – | 20 (54%) | – | 8 (33%) |
| Total energy [Kcal/day, median (q1, Q3)] | n = 162 1908 (1594,2364) | n = 120 1800 (1450,2310) | n = 40 1980 (1670,2420) | n = 27 1800 (1380,2230) | n = 20 2250 (1600,2760) |

Detailed cohort characteristics (exposures and FFQ) can be found in Supplementary Dataset 1.

*p < 0.05 CD vs. Ctl in Israel; ^p < 0.05 CD vs. Ctl in urban China; #p < 0.05 Rural vs. Urban. Most analyses were done within the rural population after we stratified subjects living in a rural area to those spending <50% of their time in the last year in urban areas ("rural", n = 88) and compared them to those living in a rural area but spending ≥50% of their time in an urban environment ("rural-urban", n = 74), and were adjusting for age in the indicated analyses.

each FFQ (see "Methods"). PCA PC1 and PC2 showed a significant correlation with the amount of protein, fat, carbohydrate, and added sugar consumption in the Chinese subcohort, with some separation of the rural controls based on PC1 and separation of urban controls based on PC2 (Fig. 3a). PCA of the Israeli sub-cohort mainly separated CD and controls on PC2, and factors contributing to this separation included fiber consumption in the direction of controls, and protein, processed meat, and added sugar in the CD direction (Fig. 3b). The diet-based correlation heatmap indicated some trends between different feature consumption in both sub-cohorts (Supplementary Fig. 2). For example, fibers and vegetables showed a high correlation with each other in both cohorts, and dietary iron (ferrous) and protein consumption also showed a high correlation, which was stronger in the Chinese cohort.

PERMANOVA was used to quantify the contribution of different factors affecting the gut microbial composition after controlling for age and gender and this was performed separately within each group in each site (rural, rural-urban, urban and CD in China, and controls and CD in Israel, Fig. 3c shows factors that were significant in at least 2 groups (i.e., independent validation), and the full list of features is in Supplementary Dataset 4). Significant factors included the amount of total and saturated fat, fruits, iron (ferrous), dairy, and added sugar daily consumption, and having farm animals or growing up with siblings. Fat (total and saturated fat, PUFA, MUFA) consumption was more specifically linked with microbial composition in China. Interestingly, higher fat intake was previously associated with CD only in an Eastern population[6]. Multivariate analyses using MaAsLin2 and controlling for age, gender, and group (rural, rural-urban, urban) including only the 283 SYS controls (FDR < 0.25), we identified 16 ASVs that showed decreased abundance with higher iron consumption (Fig. 3d, e and Supplementary Fig. 3), including *Lachnospiraceae* and *Ruminococcaceae* taxa, and 12 ASVs that were higher with increasing iron consumption, including *Actinomyces* and *Streptococcus* taxa. The heatmap shows these ASVs across samples sorted by the amount of consumed iron within each control sample group in China, and we included in the visualization also the CD group. Using MaAsLin2 we identified 44 ASVs that were reduced upon higher total fat consumption using a similar analysis, including several *Oscillospira* and *Lachnospira* taxa, as opposed to 14 ASVs that increased with higher fat consumption including two taxa from the *Veillonellaceae* family (*Acidaminococcus* and *Megasphaera* genus) and *Streptococcus*. dbBact-based analysis[14] of the ASVs that were lower with increasing iron or fat consumptions (Fig. 3d, e) indicated a large number of them have been observed in other experiments to be lower in CD (chi-square p = 3E−6), and that a

large fraction of ASVs that were higher with increasing iron or fat consumption were observed in other experiments in saliva samples (chi-square p = 0.005), and identification of salivary bacteria in the gut was previously noted in CD[11]. Similar analyses with ASVs linked with protein consumption (linked with iron consumption), showed consistent results (Supplementary Fig. 3).

## CD-increased taxa are enriched in mucosal biopsies

An unweighted UniFrac-based PCoA calculated on the 16S microbial data colored by CD diagnosis indicated disease as a significant factor (Supplementary Fig. 4a, d). Furthermore, both subcohorts showed significantly reduced Faith based alpha-diversity and decreased microbial health index in CD cases compared to controls[11] (Supplementary Fig. 4b, c, e, f). To capture CD-associated microbial ASVs and taxonomy we used Maaslin2 after controlling for gender and age, and for subject and sample type (stool ileal and rectal biopsies) in the Israeli cohort, and we included only the urban participants (healthy and CD) from China, as rural living was a major confounder. Resulting significant ASVs are shown in the heatmaps (Supplementary Fig. 4g, h and Supplementary Dataset 5), indicating biopsy samples tended to include more of the CD-increased taxa. Shared increased taxa included *Enterobacteriaceae* (ASV05780), *Actinomyces* (ASV08231), and *Fusobacteriaceae* (ASV15593) (Supplementary Dataset 5). Interestingly, using both Unweighted unifrac distance and Spearman correlation we show that the same subject's stool and biopsy samples were more similar (i.e., show lower distance) than samples taken from other subjects. Moreover, biopsies taken from the ileum and rectum, which are over one meter apart in adults, were significantly more similar to each other than the same patient's stool to either his own ileal or rectal biopsies (Supplementary Fig. 4i).

## Correlations between mucosal transcriptomics, diet, and metabolites

We further aimed to map and prioritize additional potential beneficial exposures, besides rural living, including metabolites and dietary targets that interact with the affected host CD ileal transcriptomics and the microbiome. We applied weighted gene co-expression network analysis (WGCNA) on the Israeli ileal transcriptomics datasets. By this, we aimed to capture genes and enriched pathways across samples, to not just identify the disease signal, but to break down the signal to disease-specific enriched pathways and cell types that were linked to disease, exposure, bacteria, and metabolomics features. Similar gene modules were applied to the Chinese ileal transcriptomics dataset. We

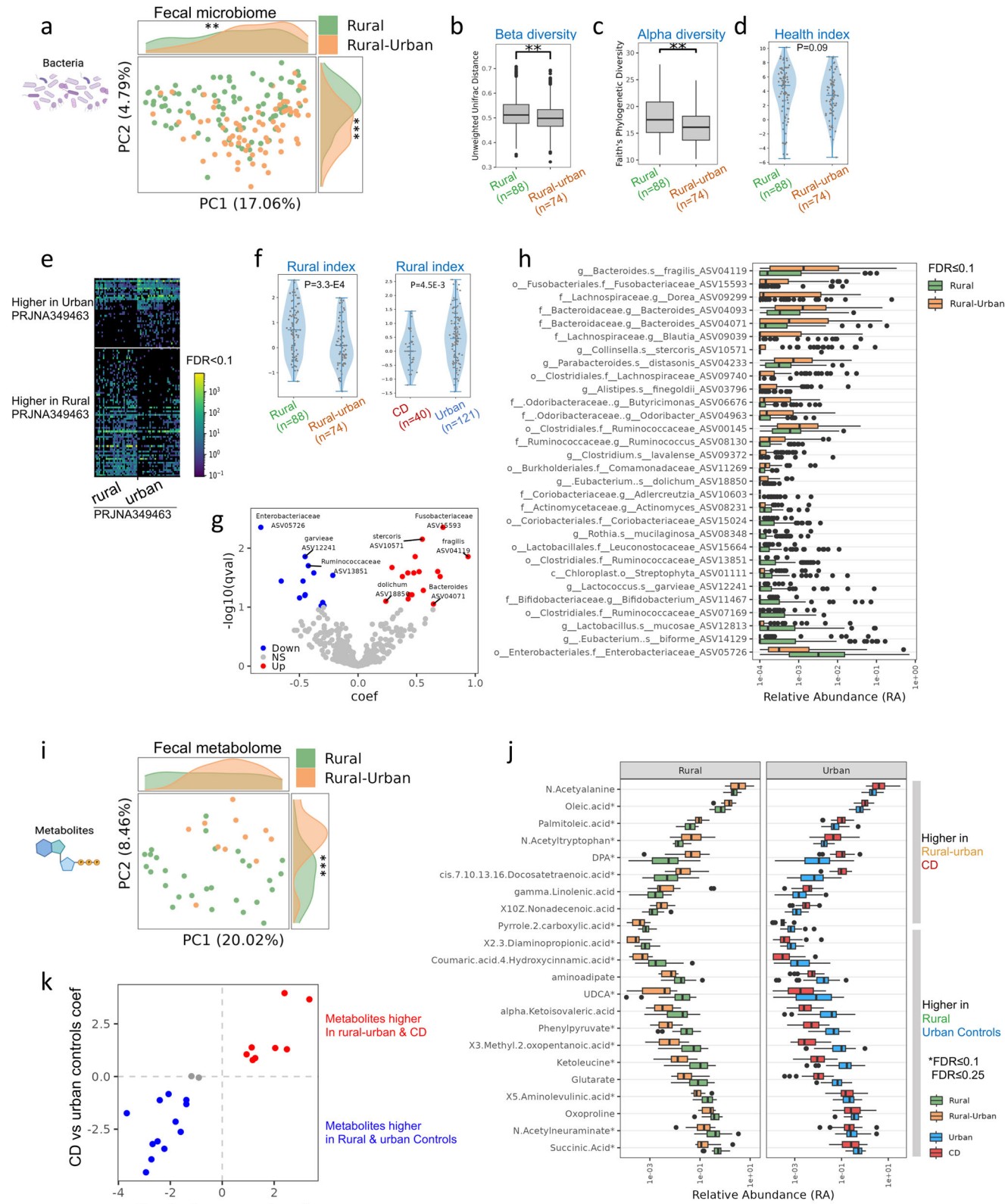

identified nine modules that were linked with CD diagnosis in Israel or China ($p \le 0.05$, each module assigned a random color as indicated in Fig. 4a, and the complete gene lists are in Supplementary Dataset 6). Functional enrichments of these modules and selected highlighted genes are shown in Fig. 4a, b. Specifically, four modules showed reduced expression in CD (heatmap colored by $r$, and the $p$ value is indicated) and were related to: (1) lipid metabolism (yellow); (2)

mitochondrial functions including translation and structure (green); (3) respiration (red); and (4) DNA damage and repair (pink). Five modules showed increased expression in CD and were related to: (1) the immune and extracellular matrix (ECM, brown module), including *CXCL* cytokines, *OSM, TREM1*, and *MMP*s; (2) myeloid signal (black module), which included TLRs, and CARD9; (3) tuft cells and eosinophils (salmon module), which included *CCR3, CLC, ALOX15* and *TFF*s;

**Fig. 2 | Gut microbial and metabolites are affected by time spent by rural residents in urban environments in China. a** Unweighted UniFrac PCoA plot of 182 rural Chinese 16S microbiome fecal samples, colored by "rural" ($n = 88$) and "rural-urban" ($n = 74$) that spends more than half of their time in urban environments. Histograms show the distribution per group on PC1/2. Unweighted unifrac distances (**b**, beta diversity, permanova $p = 0.002$), Faith's phylogenetic alpha diversity (**c**, Mann–Whitney $p = 0.004$), and our previously defined health index (**d**, Mann–Whitney $p = 0.09$) between rural ($n = 88$) and rural-urban ($n = 74$). **e** Heatmap showing ASVs with significant differential abundance between rural and urban samples (dsFDR < 0.1), using an independent cohort (BioProject PRJNA349463) from Hunan province in Southern China. Each row represents ASV and each column is a different sample. ASVs are ordered by the effect size. Those taxa were used to generate a "rural index" applied to our cohort. **f** Violin plot of rural index between rural and rural-urban (Mann–Whitney $p = 0.0001$) and Crohn Disease (CD, $n = 40$) and urban (right, $n = 121$, Mann–Whitney $p = 0.0007$). **g** Volcano plots of significantly different (FDR ≤ 0.1) taxa between rural ($n = 88$) and rural-urban ($n = 74$), using a maaslin2 controlling for age and gender (full list in Supplementary Dataset 3). **h** Boxplot showing the relative abundance of significant taxa from (**g**). **i** Canberra distance PCoA plot of 40 rural and rural-urban Chinese fecal metabolites samples colored by group, showing significant separation on PC2. **j** Boxplot showing the relative abundance of 22 significantly different (FDR ≤ 0.25, *indicates FDR ≤ 0.1) metabolites between rural and rural-urban samples, using a maaslin2 analysis controlling for age and gender (left, $n = 40$). We indicated the relative abundance in the CD and urban control groups ($n = 79$) (Supplementary Dataset 3). **k** Scatter plots of maaslin2 analysis coefficients, indicating that 20 of the 22 metabolites shown in (**j**) were also similarly significantly different in CD vs. urban controls with similar directionality (red: 8 metabolites higher in rural-urban and CD, blue: 10 higher in rural and urban controls, gray; 2 higher only in rural vs rural-urban). Two-sided *$p < 0.05$, **$p < 0.01$, ***$p < 0.001$. Boxplot center line and limit; median, upper and lower quartiles; whiskers, 1.5x interquartile range. The Violon plot center line represents the median and the kernel density estimation is in blue.

(4) innate epithelial immune functions (tan), including *DUOX2* and *CEACAM*s; and (5) cell cycle and mitosis and T and B cells (purple), which included *CCNB1, CCNE1*, and *CDK1*. Importantly, all modules were similarly altered in the Israeli and the Chinese CD cohorts, except for the DNA damage and repair (pink) and the cell cycle and mitosis and T and B cells modules (purple), which were only significantly altered in the Chinese subcohort but were changed in the same direction in the Israeli group.

We then linked these CD modules with clinical factors, biomarkers, and dietary, bacteria, and fecal metabolomics features (Fig. 4c, d). No significant associations were detected between the CD-linked modules and gender, age, BMI, or smoking in the Israeli cohort, but we identified strong associations with CRP and calprotectin (Fig. 4a). Furthermore, we additionally observed a negative correlation with our previously published microbial health index[11] which was specific to transcriptomics modules enriched for the immune system and in particular the brown (immune and ECM) and salmon (tuft cells and eosinophils) modules ($p \leq 0.01$) in the Israeli cohort. In the Chinese transcriptomics modules, BMI and gender showed no significant associations but age, smoking, ileal inflammation, and CRP were linked with the indicated modules (Fig. 4a). Significant correlations (FDR ≤ 0.25) with dietary factors in the Israeli cohort (Fig. 4c) indicated for example that starch, iodine, and selenium consumption showed similar directionality as seen with the disease, but starch was linked with the yellow epithelial metabolic functions (*APOA1* and *GASTA1*), and iodine and selenium were linked with the brown immune module (including *OSM* and *CXCL8/9*). Other examples, like Manganese and vitamin D consumption, were positively correlated with control epithelial lipid processing and mitochondrial functions, while coffee was anti-correlated with the immune modules linked with CD. Significant correlations (FDR ≤ 0.25) with dietary factors in the Chinese transcriptomics cohort were fewer and included vegetable and fruit consumption that were linked with control signals, and processed food, which was linked with the CD brown (immune and ECM) module, with trends showing correlation ($p \leq 0.05$) between vitamin D consumption and control signals, and fat consumption linked with the CD signals.

Fecal metabolites in the Israeli cohort showed a robust signal identifying 234 significant metabolites ($p < 0.05$, FDR ≤ 0.25) correlated with host transcriptomics modules (Fig. 4d and Supplementary Fig. 5a, b), of which the majority correlated with control-associated signals as indicated (Fig. 4e). Metabolites classified with lipid fatty acid classification were linked with the control-associated yellow modules, while metabolites that were classified as amino acids were linked with tan and brown CD-associated immune modules. Similar associations between metabolites and transcriptomics modules were also reflected within only the CD group (Supplementary Fig. 5b, 97.5% showed the same direction, with binomial test $p < 2.2e{-16}$). Specific potential beneficial associations included those relating to the dopaminergic system, aligning with previous studies in IBD preclinical models showing the beneficial effects of this pathway[15,16]. Specifically, phenylethanolamine was associated with the control epithelial yellow lipid metabolism module, and anti-correlated with the disease-associated black (myeloid) module, which included also S100A8 calprotectin and TLR2/8, and L-dopa was associated with control epithelial yellow lipid metabolism module, and anti-correlated with the disease-associated salmon (tuft cells and eosinophils) module. Of the 92 metabolites that were shared between the Israeli and the Chinese datasets (see "Methods"), 39 showed correlations within the TI SYS and Sheba transcriptomics gene modules. Remarkably 32 of the 39 (82%) showed similar directionally with CD and control signals in both cohorts, providing independent validation of the association seen in the Israeli cohort despite the Chinese cohort having substantially different genetics and exposures (Fig. 4f and Supplementary Fig. 5c, Supplementary Dataset 6, binomial $p = 7.1E{-5}$, see "Methods"). These 32 metabolites included tryptophan, acetyl-tryptophan, and docosatetraenoic acid positively linked with modules higher in CD, while adipate, azelate, and indole 3 methyl acetate were positively linked with control modules, and opposite from CD.

We then applied HAllA analysis[12] to capture significant associations of metabolites that were linked with each ileal transcriptomics module to dietary and microbial datasets in the Sheba cohort, as this subset had a more detailed FFQ (Supplementary Fig. 2). This showed that diet-linked fecal metabolites were associated with host epithelial metabolic functions, while microbial-linked metabolites were associated with the host immune response (Fisher's test $p < 0.00001$ for FFQ vs. 16S and MGX, Fig. 5a and Supplementary Dataset 8). Sanky plots highlighted significant positive associations between the pink transcriptomics module-associated metabolites, and dietary exposures (Fig. 5b), indicating interactions between fat and sugar consumption and fecal histamine, sucrose, and lactose. Significant positive associations between the black myeloid, brown immune and ECM, and salmon eosinophils enriched transcriptomics modules, metabolites, and metagenomics datasets (Fig. 5c, d) highlighted CD-associated metabolites (light pink). For example, methionine, which was recently shown to be crucial for myeloid cells[17], was linked with *Veillonella dispar* taxa and the black myeloid enriched transcriptomics modules. Metabolites linked with the brown immune and ECM enriched module included tyramine, which was previously shown to be secreted by *E. coli* in vitro[18]. Several mitochondria-associated metabolites were linked with CD-microbial signals: malonate, a mitochondrial respiratory-chain inhibitor, was linked with *H. parainfluenza* and *Veillonella pravula*; 3-Hydroxymethyl glutaric acid was linked with *Veillonella pravula*, and ureidopropionate was linked with *R. gnavus*. We further connected these transcriptomics-associated modules with the metagenomics-assigned pathways and enzymes classes (ECs) (Supplementary Figs. 6 and 7 and Supplementary Dataset 8).

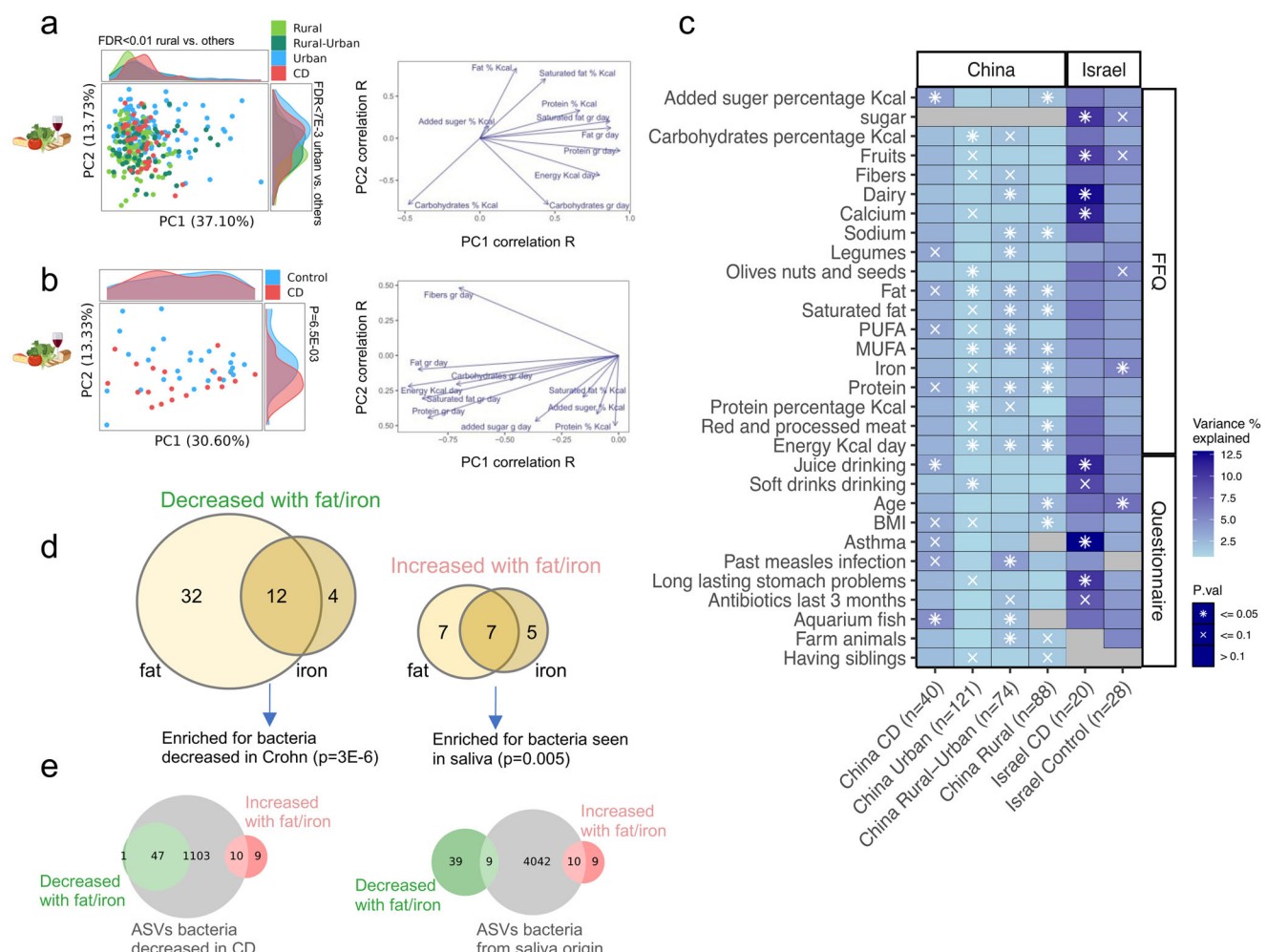

**Fig. 3 | Exposures and diet are linked with microbial variations. a** PCA figure based on food frequency questionnaire (FFQ) data and showing variations of the different groups in China (28 FFQ features, sample $n = 308$). Colors indicate the specific groups. Histograms show the distribution of samples and groups on PC1 and PC2. (right). FFQ components were correlated to the PCAs PC1 ($x$ axis) and PC2 ($y$ axis) values (left). Spearman's rho values are shown as the head of the arrow for the top 10 FFQ components with the highest PC1 or PC2 rho values. **b** Same as (**a**) for the Israeli cohort (62 FFQ feature, sample $n = 47$). **c** PERMANOVA analysis of 16S microbial variance explained by FFQ and questionnaire data, using each group and sub-cohort separately ($n$ is shown in brackets). $x$ indicates two-sided $p \le 0.1$, * indicates two-sided $p \le 0.05$. Only variables that were significant in at least two groups are shown here (full list in Supplementary Dataset 4). **d** Venn plots showing the overlap between the taxa decreased in fat and iron, and the taxa increased with

fat and iron from Supplementary Fig. 3a. **d** dbBact term associations for the fat/iron associated ASVs. **e** Number of ASVs observed (in dbBact experiments) to be associated with saliva or lower abundance in Crohn Disease (CD, right and left sub-panels respectively). Red circles show the set of ASVs positively correlated with iron and/or fat consumption, and green circles show the ASVs negatively correlated with iron and/or fat. Gray circles show the total number of dbBact ASVs associated with the term, with overlaps showing the subset of ASVs from each group that is associated with the term. 47/48 of negatively correlated ASVs have been associated at least once with decreased frequency in CD, compared to 10/19 in the positively correlated ASVs (Two-sided $p = 3E-6$, chi-square test). Similarly, 9/48 negatively associated ASVs compared to 10/19 positively associated ASVs have been observed in dbBact saliva samples (Two-sided $p = 0.005$, chi-square test). MUFA monounsaturated fatty acid, PUFA polyunsaturated fatty acid.

Metabolites associated with the control signal and the green (mitochondria) module (Supplementary Fig. 7b, c) were associated with thiamin (vitamin B1), which serves as a cofactor for mitochondrial enzymes, folate, coenzyme A, which transports fatty acids in the cytosol to the mitochondria and a precursor of the Krebs tricarboxylic acid (TCA) cycle, and flavin, which is involved in electron transfer processes. Metabolites associated with the tan (epithelial innate defense) module (including *DUOX2 & CEACAM5/6*, Supplementary Fig. 7d, e) contained several Docosatetraenoic (ω-6), Docosapentaenoic, and Eicosatrienoic polyunsaturated fatty acids (PUFA), which are interlinked with NAD+ diphosphatase, adenylate cyclase, and Enoyl-CoA hydratase, which is essential in lipid beta-oxidation.

Finally, we examined overall associations between the different datasets beyond individual features using sparse Partial Least Squares (sPLS) regression[19] (see "Methods") to maximize the shared variation between pairs of omics while accounting for the sparsity of the data in

the Sheba cohort. We identified a significantly high degree of shared variation between most omic pairs (Fig. 5e, f and Supplementary Dataset 8), and in particular between the host transcriptomics and metabolomics (0.867, $p < 0.01$), species composition and metabolomics (spearman correlation 0.848, $p < 0.01$, permutation-based analysis), and species composition and various bacterial functional profiles (0.569–0.743). A high degree of shared variation was also found between FFQ data and both functional metagenomics (0.764, $p < 0.05$) and host transcriptomics (0.65, $p < 0.08$). To integrate the different omics while accounting for the shared variation and their association with CD, we applied DIABLO[20]. This analysis demonstrated high shared explained variance between the different omics, ranging from 13%–53% (metabolomics 24%, metagenomics 13%, transcriptomics 53%, and food frequency questioners 22%) and showed an overall separation of CD from controls based on each omic (Fig. 5g). Moreover, focusing on the features most highly correlated

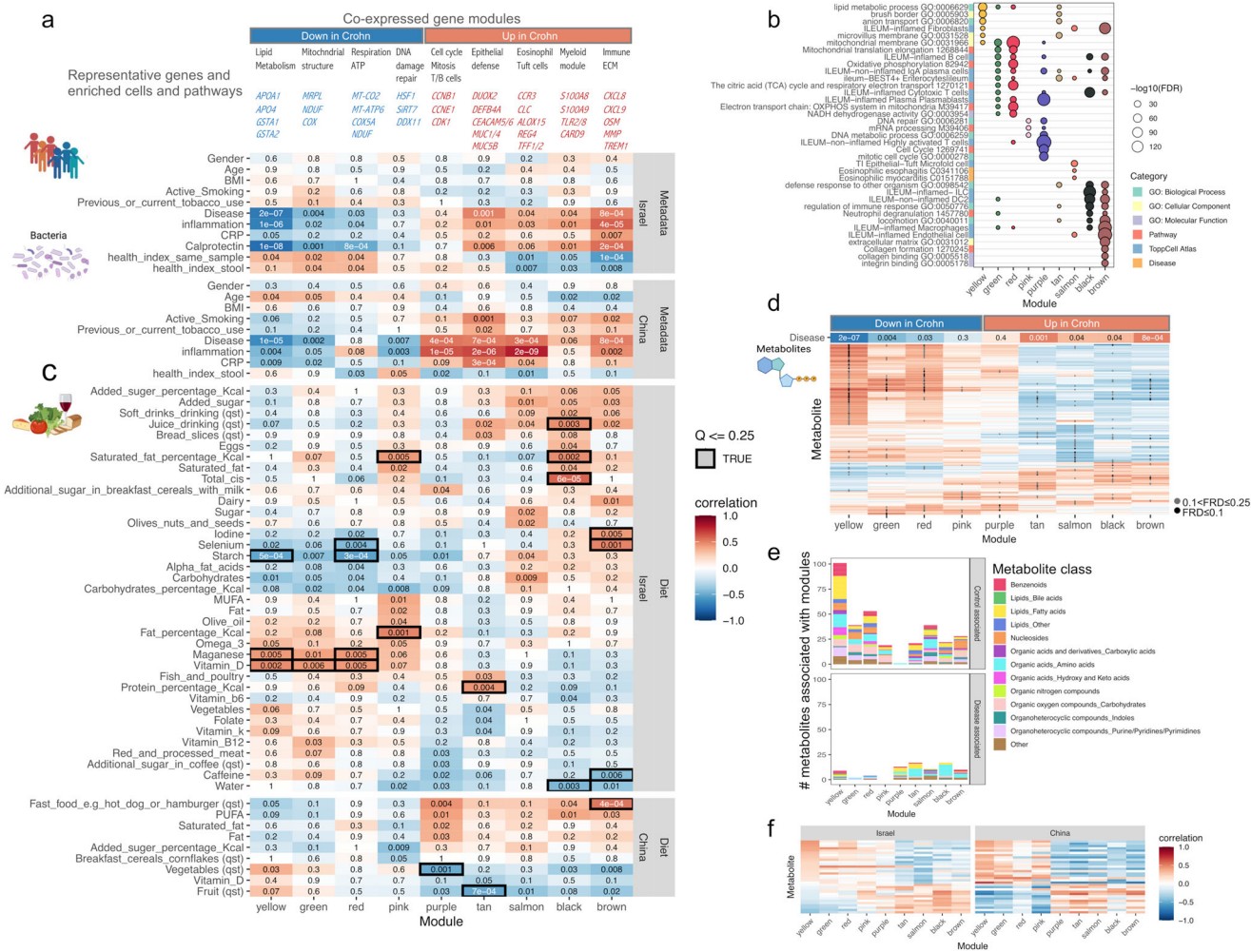

**Fig. 4 | Specific dietary factors and metabolites show correlations with Crohn disease (CD) ileal mucosal transcriptomics signals.** WGCNA co-expression modules based on the Israel ileal (*n* = 41) and applied to the China transcriptomics (*n* = 40). Modules that were correlated with Crohn disease (CD) (*p* ≤ 0.05) in either Israel or China are shown. **a** 4 modules showed reduced and 5 were induced in CD. For each module, representative genes and enriched cells/pathways are marked. Heatmap represents the correlation between each module and different features; numbers represent the correlation *p* value, and color the coefficient for each comparison. **b** ToppFun functional annotation enrichment of genes within each module. FDR is shown as the circle size; manually selected annotations origin database is marked on the *y*-axis (full list in Supplementary Dataset 6). **c** Heatmap of the correlations between each module and dietary factors, with numbers representing the correlation *p* value and color for the coefficient. Only factors with *p* ≤ 0.05 (two-sided) in at least one module are shown, and correlations with Benjamini–Hochberg FDR ≤ 0.25 are marked with a black square. **d** Heatmap of the correlation between each module and stool metabolites, colored by correlation

coefficient. Only metabolites with Benjamini–Hochberg FDR ≤ 0.25 in at least one module are shown, and those significant correlations (two-sided *p* < 0.05 and FDR ≤ 0.1, or *p* < 0.1 and FDR ≤ 0.25) are marked with black and gray dots respectively. **e** Bar graph showing the number of metabolites significantly correlated with each of the modules, separated by CD- or control-associated correlations defined by the direction of the metabolite-module correlation and the direction of the module compared to disease. The colors represent the metabolites class based on Human Metabolome Database (HMDB). **f** Heatmap of the correlation between each module and the 32 stool metabolites (of 91 common metabolites in the SYS and Sheba datasets) that showed significant correlation in the same direction with CD in Israel and China, colored by correlation coefficient (detailed heatmap in Supplementary Fig. 5c, full list in Supplementary Dataset 7). Metabolite's direction is defined as the direction of the strongest correlation between all the modules. MUFA monounsaturated fatty acid, PUFA polyunsaturated fatty acid, ECM extracellular matrix.

with disease state (top 10 loadings from each omic across components 1 and 2), we cataloged 67 features across omics that were associated with CD (Fig. 5h and Supplementary Dataset 8). Notably, these features include *R. gnavus* and *E. Ramosums*, which clustered with the brown (immune and ECM) module that included CXCL cytokines, OSM, TREM1, and MMPs, and the back (myeloid) module, which included TLRs, and CARD9, transcriptomics modules in the right CD space, which is consistent with the results in Fig. 5c, and on the opposite side from features like fibers, vegetables, and *A. putredinis*, which clustered at the controls space on the left (an interactive map of all prioritized features in Fig. 5h can be found as Supplementary Dataset 9).

## Discussion

Understanding the staggering rise in CD incidence in increasingly industrialized communities is still challenging. In this study, we exploited a consortium uniquely set to interrogate profiles in urban and rural communities in a developing country (China), with its heavily modernized cities juxtaposed to still underdeveloped rural communities and compared those with a Westernized Israeli population. This design enabled a smoother comparison between rural-urban signals and the CD signal in China. We showed that gut microbial and metabolomics changes observed in the rural-urban transitional group mirrored those seen in CD. Other advantages include the use of newly diagnosed and treatment naïve CD patients and controls, the analysis

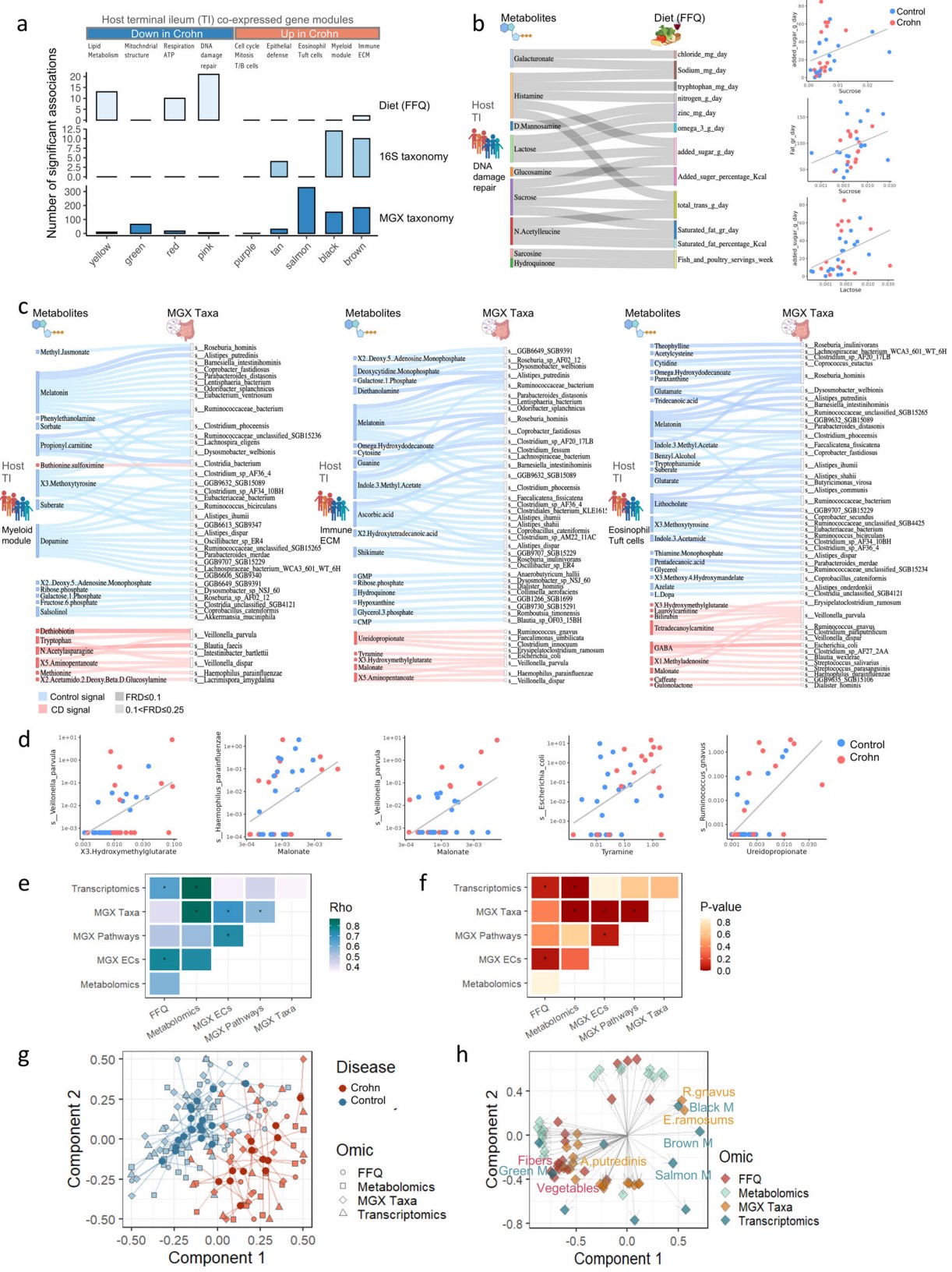

of paired personal dietary exposure and omics within each subject, and the ability to independently validate the signal between the Chinese and Israeli sub-cohorts. Characterizing the different omics layers and dietary exposures directly in each subject enabled us to capture interactions between diet, host, microbiome, and metabolites and develop integrated analyses of CD pathogenesis on the health-disease

axis. This diet and omics dataset which includes subjects living outside of North America can be used as a valuable resource for the larger community to generate hypotheses linked to different exposures. It supplements other CD omics studies from North America, including our previous transcriptomics and microbiome analyses in the RISK study[7,8,21,22], which did not include diet and metabolomics and Human

**Fig. 5 | Host epithelial-linked metabolites were associated with diet while host immune-linked metabolites were correlated with the microbiome.** The correlations between metabolites associated with each ileal transcriptomics module in the Israeli subcohort (n = 41) were tested against FFQ and microbiome data separately using HAllA (Hierarchical All-against-All Association Testing) with p < 0.05, FDR ≤ 0.25. **a** Bar plot of the number of significant correlations between metabolites associated with the different modules and food frequency questionnaire (FFQ, n = 33), 16S taxonomy (n = 36), and metagenomics (MGX) taxonomy (n = 37). Full lists in Supplementary Dataset 8. **b** Sankey figure shows significant correlations between pink module-associated metabolites and FFQ. On the right are scatterplots of 3 example metabolites and FFQ components. **c** Sankey figure of up to top 50 significant correlations for control-associated metabolites (blue) and disease-associated metabolites (red), for metabolites associated with black (left), brown (middle), and salmon (right) modules, and MGX taxonomy. Only positive correlations are shown. **d** Scatter plots of example metabolites and MGX species correlations. **e**, **f** Pairwise sparse Partial Least Squares (sPLS) regression between fecal metabolomics, host transcriptomics WGCNA modules, FFQ, MGX taxonomic, and functional profiles (pathway and ECs). Spearman correlation between the first sPLS components the defined two omics (**e**) and p values calculated based on shuffled data (**f**) are shown. Pairs with two-sided p < 0.1 are marked with an asterisk. **g** Ordination DIABLO analysis according to omics (shape) and disease state (Crohn−red; healthy−blue). Each sample is described by 4 omics, FFQ components, metabolomics, metagenomics, and host transcriptomics PC1. Sample centroids are plotted in bold solid dots and connected by a line to all omics measurements. **h** Correlation circle plot for the DIABLO analysis, where each point represents a molecular feature. The point position is defined according to its correlation with the first and second components. Only MGX taxonomy is included as the MGX EC and pathway profiles were highly correlated with the taxonomy profiles. An interactive version of this plot can be found in the Supplementary Information in Supplementary Dataset 8 and 9. CD Crohn disease, ECM extracellular matrix, MGX metagenomics, TI terminal ileum, ECs enzymes classes.

Microbiome Project 2 (HMP2) IBD that included more limited dietary exposures[23,24]. In addition to the suggested signal seen in the rural-urban transitional group in China, our analyses further map and prioritize potential beneficial dietary targets and metabolites that interact and negatively correlate with the CD host ileal transcriptomics and/or the microbiome and may be used to redirect patients to healthier states.

We applied an unbiased approach [weighted gene co-expression network analysis (WGCNA)[25]] to the mucosal ileal transcriptomics data to define different gene co-expression modules with enriched functional pathways. Using this approach enabled the separation of the disease signal into different and more specific enriched functional modules. The identification of these per-subject signals of the transcriptome module with specific metadata features including diet, metabolomics datasets, and gut microbiome. Interestingly, diet-linked fecal metabolites were associated with host ileal epithelial metabolic functions, while the microbe-linked metabolites and the summarized microbial health index were associated with ileal transcriptomics modules enriched for immune genes and functions. Analysis of the associations between transcriptomics and diet indicated, for example, that higher consumption of manganese, vitamin D, and coffee is negatively correlated with signatures seen in the disease, and positively correlated with a healthier microbial composition. Numerous epidemiological, laboratory, and clinical studies have already demonstrated the protective role of vitamin D in the development and course of CD[26], and its supplementation is part of the therapeutic recommendation. The roles of manganese and coffee are less established, although previous data already alluded toward the potential benefit of coffee in different models[27,28] and manganese[29,30] was shown to be an important factor for proper maintenance of the intestinal barrier and provides protection against DSS-induced colon injury. In contrast, consumption of sugar and saturated fat was positively correlated with the transcriptomic signature seen in CD. Analysis of the associations between transcriptomics and metabolomics highlighted 234 metabolites that show potential beneficial effects to drive the CD signal toward a healthier state, of which a subset was also validated in the SYS dataset. One promising example is azelate, which is currently used to suppress inflammation in the skin, likely through the activation of PPARgamma, which is also relevant in the gut[13]. Other potentially beneficial metabolites are related to the dopaminergic system (phenylethanolamine & L-dopa), aligning with previous studies in IBD preclinical models showing the beneficial effects of this pathway. Interestingly, a prominent inhibition of mitochondrial functions was already defined in IBD[31–34], and here we highlight associations between metabolites that are known to suppress mitochondrial respiration and functions such as malonate and 3-Hydroxymethyl glutaric acid with several CD-associated bacteria including *Veillonella pravula*, *R. gnavus*, and *H. parainfluenza*, supporting a link between mitochondrial

inhibition and microbial signals. A complementary multi-omics approach supported these findings; sPLS regression identified a significantly high degree of shared variation between most omic pairs and between the host transcriptomics and metabolomics (0.867, p < 0.01) and FFQ data (0.65, p < 0.08), and integration of different omics while accounting for the shared variation and their association with CD highlighted features including *R. gnavus* and *E. Ramosums*, which clustered with the immune and ECM and the myeloid transcriptomics modules, and opposed features like fibers, vegetables, and *A. putredinis*, which clustered at the controls space.

Epidemiological studies have highlighted that increased consumption of certain Western dietary components, such as processed meat, saturated fats, and starchy desserts, is linked with an increased risk of CD[35]. Here, we analyzed a heterogeneous Asian population from rural and urban China and a second cohort from Israel composed of a Westernized White population. We demonstrated that exposure and diet, including total fat and iron consumption, are significantly linked with microbial variations. We found that taxa that were decreased with increasing iron or fat consumption were enriched for taxa previously shown to be decreased in CD, and those taxa that were higher with increasing fat and iron were enriched for bacteria usually seen in saliva samples, also previously linked with CD[7,11]. Another interesting observation involved individuals who live in rural areas but spent a significant fraction of their time (above 50%) in urban area ("rural-urban" participants). Compared to rural participants, rural-urban cases had higher flush toilet availability, lower exposure to farm animals, and increased exposure to soft drinks and coffee in the rural-urban group. Faith-based Alpha-diversity was lower in rural-urban vs. rural controls, and in CD vs. urban cases. Our previously defined microbial health index[11] was also significantly reduced in the rural-urban individuals vs. rural controls, and an independent rural index was lower in rural-urban vs. rural controls and was lower also in CD vs. urban controls. MaAsLin multivariate analyses identified a higher abundance of several *Bacteroides*, *R. gnavus*, and *Fusobacteriaceae* taxa, which were previously linked with CD, in rural-urban compared to rural cases. Metabolomics analyses also indicated that significant changes seen in this rural-urban group significantly overlapped with changes also captured in CD. These findings indicate that rather than the conventional dichotomous classification often employed to interrogate rural versus urban residents' biology, the time spent by remote rural residents in urban environment is a significant driver of microbial profiles potentially linked with exposure and diet, indicating a continuum rather than an on/off effect of the exposome. The similarities observed in the metabolites and gut microbiome in rural-urban transition and in CD, may suggest the contribution of these factors to the pathogenesis of this complex and multifactorial disease.

Our study has several strengths. We heavily relied on analyzing human cohorts and biospecimens to capture pathogenesis and

diet-host-microbial interactions since there are no available CD murine models that fully recapitulate human aspects of the disease. We included several omics domains and detailed dietary and environmental exposure and performed correlations after identifying specific ileal gene expression modules that are not only related to CD but also are enriched for specific pathways and cell types. The results obtained can be translated into the manipulation of top-incriminated dietary and environmental triggers or by novel agents designed to specifically target interactions at critical junctures. Moreover, we included paired metabolomics datasets on a subset of samples. The metabolome is a central downstream output, mediator, and effector comprised of derivatives of the microbiome, host, environment, and diet; factors that are all linked with CD. Since metabolites are chemical compounds, they can conveniently be applied and tested as potential interventions, and we mapped and prioritized these with potential beneficial signals that are linked with a healthier state, and were able to validate a group that was also present in the Chinese dataset independently. Finally, we included a unique population in China of a rural underdeveloped community 300 km north of Guangzhou, and stratified these rural residents by the time they spent in urban environments. This highlighted changes that mirror gut microbiome and metabolite changes seen in CD.

Limitations of our work include performing multi-omics analyses on a relatively small cohort covering two geographically very different sites and the more limited metabolomics and transcriptomics data, which was only available for a subset of cohorts. It is possible that with an enlarged cohort in the indicated ethnicities, we could have identified other weaker signals that we were not able to detect with the current cohort size. However, using the current cohort, we were able to find important and substantial results using rigorous analyses. The dietary information in our study was obtained using FFQs, which are frequently used to monitor dietary habits but have limitations. While more accurate participant dietary information could potentially lead to the detection of additional signals that did not reach statistical significance in our analysis, we were still able to detect diet-associated changes with significant host-omics effects. In addition, as each group in our cohort originates from a single location, it may affect the generalizability of the results to other locations. There is a general tradeoff between achieving higher sensitivity by taking a more homogenous population, and better generalizability by including a more heterogenous population. We opted for a design comprised of two countries, and when possible, used them as independent validation, as well as using the rural index derived from another independent study, hence increasing the generalizability of some of our main results. Another cohort-related limitation is that since CD is rare in rural areas, our cohorts did not include analyses from this subgroup and CD patients were from urban areas. We used throughout FDR cutoff of 0.25, but we supplemented the results, when possible, with independently validated, we indicated the features that had FDR cutoff of 0.1, and *p* and FDR values are given in the suppl. datasets. It is possible that with an enlarged cohort in the indicated ethnicities, we could have identified other weaker signals that we were not able to detect with the current cohort size. However, using the current cohort, we were able to find important and substantial results using rigorous analyses. We did not include single-cell transcriptomics, but there are advantages to using whole mucosal gut biopsies, which are used for diagnosis and follow-up in the clinical setting. Enteric infection burden was not directly tested, but CRP levels that indirectly indicate potential infections were not different between rural and rural-urban (Supplementary Dataset 1). Additionally, due to limitations associated with the COVID-19 outbreak, samples from China and Israel were processed locally within each country of origin, but both cohorts were analyzed using a similar pipeline and as in the case of the metabolomics association with the transcriptomics modules were used as an independent validation.

In summary, this SOURCE multi-omics cohort included datasets and samples from diverse populations from both East (China) and West (Israel) and enabled thorough exploration of diet-omics domains simultaneously to facilitate the detection of complex interactions in the gut. Our human-based data has identified interesting associations and can direct toward potential beneficial metabolites and dietary modifications that can be further tested in future interventional studies in models and/or human studies.

## Methods

### Study population
The study was approved by ethics committees in both Sheba Hospital in Israel (No. 5484) and the First-affiliated hospital of Sun Yat-Sen (SYS) University in China (No.[2021]GH1457; No.[2019]GH0367; No.[2017]073), and the research complies with all relevant ethical regulations. This study was conducted in Guangdong province in Southern China and Israel (Jan 2019 to April 2021). Several populations were analyzed (Fig. 1). In China, newly diagnosed and treatment-naïve CD patients were included, along with healthy urban residents of Guangzhou, a modernized metropolitan community with a population of 16 million, and healthy residents of Shaoguan district, a rural underdeveloped community 300 km north of Guangzhou. Participants were asked about the amount of time they spent in an urban environment in the last year ("How long have you stayed in a city in the last year?") with answers including less than 10%, 10–50%, and above 50%. Newly diagnosed CD patients and healthy controls from Israel were included as another layer of a Westernized control cohort. CD diagnoses of all patients were harmonized and based on clinical history, physical examination, laboratory work, radiological findings, and endoscopic and histological features as previously established in the European Crohn's and Colitis Organization consensus statement[36]. Written informed consent was obtained from all participants.

### Data collection and biospecimens
Data regarding enrolled subjects were recorded in a structured manner that included demographic, clinical, laboratory, endoscopic, and pathological features for the indicated group and participants (Fig. 1). Laboratory tests included C-reactive protein (CRP). Endoscopic evaluations included gut segmental involvement. Stool specimens were collected into a collection tube at least 3 weeks following any antibiotic treatment. Stool samples were aliquoted and frozen immediately in −80 °C. Ileal biopsies were gathered during diagnostic colonoscopy and stored in RNAlater and frozen at −80 °C. Due to COVID-19 outbreak limitations, samples were processed locally within each country of origin. Samples from both cohorts were analyzed using a similar bioinformatic pipeline; direct comparisons between groups were performed within each country of origin.

### Environmental and dietary questionnaires
Patients from both cohorts underwent environmental and dietary exposure surveys. Gender was determined based on self-report. For environmental exposure, we used the questionnaire developed by the International Organization of IBD (IOIBD), with some modifications. The questionnaire consists of 87 questions covering 25 different topics proposed to be environmental risk factors for CD. Although it was not formally validated by the IOIBD, this questionnaire has been previously used in epidemiological studies investigating triggers of IBD[37,38], including one conducted in South-East Asia and China[39]. IOIBD questions relate to five main different areas: (1) Childhood factors up to 20 years including breastfeeding, appendectomy, tonsillectomy, eczema, vaccinations (tuberculosis, pertussis, measles, rubella, diphtheria, tetanus, polio), childhood infections (measles, pertussis, rubella, chickenpox, mumps, scarlet fever) and pet ownership; (2) food habits including daily, weekly or less frequent consumption of fruit, vegetables, egg, cereal, bread, coffee, tea, juice, sugar, and fast food;

(3) smoking habits (current smoker, non-smoker, ex-smoker); (4) sanitary conditions such as the availability of in-house water tap, hot water tap or flush toilet; and (5) others factors including daily physical activity, the oral contraceptive pill and stressful events before diagnosis. We also included in the IOIBD questionnaire items about antibiotic use before and after the age of 15 years, use of toothpaste, and the presence of amalgam teeth filling during childhood or later in life. In addition, we added a specific question specifying the amount of time spent in an urban environment in the last year with answers including less than 10%, 10–50%, and above 50%. Although the IOIBD questionnaire explores the role of diet before the diagnosis of IBD, we added an additional comprehensive FFQ (Food frequency questionnaire), conducted by a trained dietician. This tool included over 600 food items, with an FFQ list that prioritized the foods/beverages accounting for at least 80% of the total energy intake and the between-person variance in previously collected dietary intake data from the adult Israeli population[40,41]. Personnel from both Sheba and SYS were similarly trained in the dietary interview method and the equivalent FFQ was used after translation to Chinese and adaptation to the Chinese diet. In Israel, a computerized FFQ version was used that automatically computes the average daily nutrient content of an individual patient's diet including macro and micronutrients, and food item servings. In China, the data were extracted manually and the portion of a specific food, and micro and micronutrient consumption were summarized. Data regarding exposures and diet are summarized in Supplementary Dataset 1. The Sheba FFQ output included 62 features, while the SYS one included 28 features.

Principal component analysis (PCA) was performed to summarize variation in Israel and China FFQ data, separately, using R prcomp function with data scaled and centered. FFQ components were correlated to PC1 and PC2 values using Spearman's correlation, and the top 10 FFQ features by maximal Spearman's rho correlation to PC1 or PC2, with $p \leq 0.05$, are shown to highlight the main components affecting variance in dietary intake. Spearman's correlation was calculated between all FFQ component pairs, separately for Israel, China rural, and China urban and CD. Correlation heatmaps were generated showing Spearman's rho values, clustered using R hclust function with Euclidian distances.

## Fecal DNA extraction and 16S amplicon sequencing

At the Sheba site (Israel), fecal[42–44] and biopsy[7–10] DNA extraction, and PCR amplification of the variable region 4 (V4) of the 16S rRNA gene using Illumina adapted universal primers 515F/806R was conducted using the direct PCR protocol [Extract-N-Amp Plant PCR kit (Sigma-Aldrich, Inc.)][42–44]. At the SYS site (China), fecal DNA samples were extracted using the OMEGA Soil DNA Kit (Omega Bio-Tek, Norcross, GA, USA) following the manufacturer's instructions. PCRs of the variable region 4 (V4) of the 16S rRNA were conducted and amplicons were pooled in equimolar concentrations into a composite sample that was size selected (300–500 bp) using agarose gel to reduce non-specific products from host DNA. Sequencing was performed on the Illumina MiSeq platform at Sheba or the NovaSeq platform at Shanghai Personal Biotechnology Co., Ltd (Shanghai, China). Samples from both cohorts were analyzed independently using a similar bioinformatic pipeline. Reads were processed in a data curation pipeline implemented in QIIME 2 version 2021.4[45,46]. Reads were demultiplexed according to sample-specific barcodes. Quality control was performed by truncating reads after three consecutive Phred scores lower than 20. Reads with ambiguous base calls or shorter than 150 bp after quality truncation were discarded. Amplicon sequence variants (ASVs) detection was performed using Deblur[47], and duplicate samples from different runs were joined, resulting in 323 samples with a median of 55,844 reads/sample (IQR 50,435–62,947) for China, and 158 samples with a median of 23,595 reads/sample (IQR 13,549–36,505) for Israel. ASVs present in less than 1% of the samples were removed.

Additionally, candidate contaminant ASVs were filtered using dbBact[14] by removing ASVs with the $f$-score mean for ("water", "soil", "mus musculus") higher than "homo sapiens", resulting in 1668/3642 ASVs for China and 1290/2838 ASVs for Israel after filtering. ASV taxonomic classification was assigned using a naive Bayes fitted classifier, trained on the August 2013 Greengenes database. Taxonomy assigned by 16S is indicated by the specific ASV number, and the sequence associated with each ASV number is indicated in Supplementary Dataset 2 and in the relevant supplementary datasets. All samples were rarefied to 33k reads for α and β diversity analysis for China samples, and 4k reads for Israel samples, to avoid read number effects. Faith's phylogenetic diversity[48] was used as a measure of within sample α diversity, and Unweighted UniFrac was used as a measure of between sample β-diversity[49], using a phylogenetic tree generated by SATé-enabled phylogenetic placement (SEPP)[50]. The resulting distance matrix was used to perform a Principal Coordinates Analysis (PCoA). Heatmaps were generated using Calour version 2018.10.1 with default parameters[51].

PERMANOVA: quantifications of variance were calculated using PERMANOVA with the adonis function in the R package Vegan[52], on the rarefied Unweighted UniFrac distance values. The total variance explained by each variable was calculated while accounting for age and gender in the model (except for when looking at the contribution of age and gender, when only age or gender can be controlled for). PERMANOVA was calculated independently for each group (China CD, urban, rural-urban and rural, and Israel CD and control) and for each questionnaire and FFQ component. Multivariate Association with Linear Models (MaAsLin2) was used with R package version 1.8.0, to test for specific differentially abundant ASVs between: rural and rural-urban samples controlling for age and gender, China CD and urban controls controlling for age and gender, and Israel CD and controls, using both stool and biopsy samples, controlling for age, gender, sample type (stool or biopsy) and patient ID as the random variable. MaAsLin2 was also used to identify ASVs correlated with dietary consumption of fat, iron, and protein separately, within all China controls, controlling for age, gender and group (urban, rural-urban or urban). A false discovery rate (FDR) cutoff of 0.25 was used for all MaAsLin2 analysis[53], and FDR cutoff of 0.1 is indicated.

Rural and Health indices: per-sample health index was calculated as previously described[11]. Briefly, a set of ASVs that were significantly increased or decreased across multiple human diseases compared to controls was identified. Using these ASVs bacteria, for each sample the log of the ratio of health-associated bacteria (98) to disease-associated bacteria (32), following rank transforming the samples, was calculated and defined as the health index (with higher values indicating a better health-associated microbiome). A similar approach was used to define a "rural index" for each sample, as follows: Using an independent dataset of rural and urban Chinese samples[5], we identified 76 and 42 ASVs significantly higher/lower in the rural community respectively (using a rank-mean test with dsFDR<0.1[54], implemented in Calour[51]). The rural index was then calculated for each SOURCE sample as the log of the ratio of the rank-transformed frequencies of the ASVs from the rural and urban ASVs. Age matching between sample groups was performed by binning ages into 10 year bins, and equalizing the number of samples in each age bin between the two sample groups by randomly dropping samples.

dbBact term enrichment analysis: significantly enriched dbBact[14] ontology terms between two ASV sets (e.g., higher/lower in rural community or positively/negatively correlated with dietary factor) were identified using the dbBact-calour plugin. Briefly, dbBact contains annotations linking ASVs to ontology-based terms, based on manual analysis of over 1000 amplicon experiments. For the current experiment analysis, for each term, a dbBact annotation-based score is calculated for each ASV, and the distribution of the score across the two ASV groups is compared to random permutations (of ASV group

labels), using a permutation-based rank-mean test with dsFDR multiple hypothesis correction. For the term-specific Venn diagrams, the number of ASVs associated with the term in at least one dbBact experiment is shown for each ASV group, with the central (gray) circle showing the total number of ASVs in dbBact associated with the term. The study was conducted and is currently reported according to the STORMS guidelines[55] (information in Supplementary Dataset 1).

## Shotgun metagenomics sequencing

For samples from the Sheba site, DNA was purified using the Power-Mag Soil DNA isolation kit (MoBio) optimized for the Tecan automated platform. DNA was diluted to 1.5 ng, and Illumina libraries were prepared using Nextera DNA library preparation kit, Ref# 15028211; by Tecan Freedom Evo 200 robot device. Nextera DNA Unique Dual Indexes Sets A–D from IDT were used for library preparation. Library concentration was measured using the iQuantTM dsDNA HS Assay Kit, ABP biosciences (Cat# AP-N011), and library size was quantified by automated electrophoresis nucleic acid QC -Tape-Station system. Libraries were sequenced by a NextSeq 500 device with IlluminaNS 500/550 High Output V2 75 cycle kit, Cat# FC-404-2005. SYS samples site, the extracted DNA was processed to construct metagenome shotgun sequencing libraries with insert sizes of 400 bp by using Illumina TruSeq Nano DNA LT Library Preparation Kit. Each library was sequenced by the Illumina HiSeq X-ten platform (Illumina, USA) with PE150 strategy at Personal Biotechnology Co., Ltd. (Shanghai, China). Samples from both cohorts were analyzed independently using a similar pipeline (https://github.com/biobakery). Reads were first decontaminated and trimmed using KneadData v0.12.0, then samples under 7M reads PE (or 3.5M SE) were excluded. The average number of reads after the quality control process was 52,508,783.72 (±11,833,357.89) PE and 8,773,250.37 (±2,135,013.94) SE for the Chinese and Israeli cohorts respectively. Taxonomic profiles were generated using MetPhaln v4.0.0[56], from which the functional profiles were generated using HUMAnN v3.6[57]. Default parameters were used for all modules, besides defining concatanating PE reads using the -cat-final-output parameter in KneadData. Taxonomic and functional features were filtered out if they didn't have abundance greater than 0.01% in at least 10% of the samples.

## Fecal metabolomics profiling and data preprocessing

For the Sheba samples ($n = 37$), extraction solution (ES: 75% methanol and 25% water and six internal standards) was mixed with fecal smear, sonicated for 10 min, centrifuged at $14,000 \times g$ for 10 min at 4 °C, and stored at −80 °C until submission for LC-MS metabolomics analysis. LC-MS analysis was conducted as described[58]. Briefly, Dionex Ultimate ultra-high-performance liquid chromatography (UPLC) system coupled to Orbitrap Q-Exactive Mass Spectrometer (Thermo Fisher Scientific) was used. The resolution was set to 35,000 at a 200 mass/charge ratio ($m/z$) with electrospray ionization and polarity switching mode to enable both positive and negative ions across a mass range of 67–1000 $m/z$. The UPLC setup consisted of ZIC-pHILIC column (SeQuant; 150 mm × 2.1 mm, 5 μm; Merck). Stool extracts were injected, and the compounds were separated with mobile phase gradient, starting at 20% aqueous (20 mM ammonium carbonate adjusted to pH 9.2 with 0.1% of 25% ammonium hydroxide) and 80% organic (acetonitrile) and terminated with 20% acetonitrile. Flow rate and column temperature were maintained at 0.2 ml/min and 45 °C, respectively, for a total run time of 27 min. All metabolites were detected using mass accuracy below 5 ppm. Thermo Xcalibur 4.1 was used for data acquisition. Peak areas of metabolites were determined using MZmine2.53[59] by using the exact mass of the singly charged ions ($m/z$) and the retention time of metabolites was predetermined on the pHILIC column by analyzing an in-house mass spectrometry metabolite library that was built by running commercially available standards. Thermo TraceFinderTM 4.1 software was used for validation, by comparing the

peak areas of the internal standards determined by both software. A total of 405 of 545 of the predefined metabolites library passed the threshold of peak intensity and were included in the analyses.

For the SYS samples, a targeted metabolomic analysis using a Q300 kit (Metabo-Profile, Shanghai, China) was performed. Lyophilized samples (~5 mg) were mixed with 25 μl of water and homogenized with zirconium oxide beads for 3 min. One hundred twenty μl of methanol containing internal standard was added and then homogenized for another 3 min, centrifuged at $18,000 \times g$ for 20 min, and 20 μl of supernatant was transferred to a 96-well plate. The plate was sealed and incubated at 30 °C for 60 min, after which 330 μl of ice-cold 50% methanol solution was added to dilute the sample. Then the plate was stored at −20 °C for 20 min, followed by $4000 \times g$ centrifugation at 4 °C for 30 min. One hundred thirty-five μl of supernatant was transferred to a new 96-well plate with 10 μl internal standards in each well. A liquid chromatography coupled to tandem mass spectrometry (UPLC-MS/MS) system (ACQUITY UPLC-Xevo TQ-S, Waters Corp., Milford, MA, USA) was used to quantitate all targeted metabolites. To diminish analytical bias within the entire analytical process, the samples were analyzed in duplicates that were randomly analyzed. The quality control (QC) samples, internal standard calibrators, and blank samples were analyzed across the entire sample set. The raw data files generated by UPLC-MS/MS were processed using the MassLynx software (v4.1, Waters, Milford, MA, USA) to perform peak integration, calibration, and quantitation for each metabolite. A total of 185 of 305 metabolites passed the threshold of peak intensity and were included in the analyses.

Ninety-two metabolites overlapped between the 185 SYS dataset and the 405 in the Sheba cohort, and these were used to test correlations as independent validation. Overall, the normalized metabolite levels (each metabolite value was divided by the sum of total metabolites value per sample) were used for all downstream analyses in both Sheba and SYS cohorts.

Principal coordinates analysis (PCoA) was performed on the metabolomics data using Canberra distances as a measure of between sample β-diversity. Multivariate Association with Linear Models (MaAsLin2) was used with R package version 1.8.0, to test for specific differentially abundant metabolites between China rural and rural-urban samples controlling for age and gender, and for China CD and urban controls controlling for age and gender, using FDR cutoff of 0.25.

## RNA extraction and RNA-seq analysis

At the Sheba site, RNA and DNA were isolated from terminal ileum (TI) biopsies obtained during diagnostic colonoscopy using the Qiagen AllPrep RNA/DNA Mini Kit. PolyA-RNA selection, fragmentation, cDNA synthesis, adapter ligation, TruSeq RNA sample library preparation (Illumina, San Diego, CA), and paired-end 75 bp sequencing were performed. Median reads depth was ~ 39M (31–46M IQR) in Sheba. Samples were sequenced at the NIH -supported Cincinnati Children's Hospital Research Foundation Digestive Health Center. At the SYS site, mRNA was purified from total RNA using poly-T oligo-attached magnetic beads. The strand-specific cDNA sequencing libraries were generated using NEBNext® UltraTM Directional RNA Library Prep Kit for Illumina® (NEB, USA), and index codes were added. Samples were purified (AMPure XP system) and clustering of the index-coded samples was performed on a cBot Cluster Generation System using HiSeq 4000 PE Cluster Kit (Illumina, NEB, USA). After cluster generation, the stranded, poly-A selected libraries were sequenced on an Illumina NovaSeq 6000 platform by Novogene Bioinformation Technology. One hundred fifty bp paired-end reads were generated to a median depth of 42.3M (39.9–46.6M IQR) reads for China samples, and 38.2M (32–38.2M IQR) reads for Israel samples. Reads were quantified by kallisto[60] version 42.5 using Gencode v24 as the reference genome. Kallisto output files were summarized to gene level using R package tximport version 1.22.0[61]. Protein coding genes with Transcripts per

Million (TPM) values above 1 in at least 20% of the samples were used in downstream analysis.

## Transcriptomics WGCNA and correlations with diet and metabolomics

Weighted gene co-expression network analysis (WGCNA) to identify modules of co-expressed genes[25,62] was implemented utilizing R WGCNA package version 1.72-1, using the Sheba cohort as described[63,64]. The analysis was performed on TI transcriptomics data. WGCNA was implemented for the identification of co-expressed gene clusters; we used pairwise correlations between gene expression profiles and the signed hybrid version of WGCNA. Similarities of gene co-expression are converted to adjacency values (power adjacency function), with a β parameter of 12. Average linkage hierarchical clustering on TOM-based dissimilarities is implemented to detect modules of strongly correlated genes across samples. For each module, the first principal component, referred to as the eigengene, was considered to be the module representative. A module summarized the expression levels of all the genes in a given module. Included parameters were cluster sensitivity parameter (deepSplit) of 2 to identify balanced gene modules, a minimum number of genes in a module (minModuleSize) was set to 30 genes, maxBlockSize was set to 20,000 to include all genes in one block. The same modules of co-expressed genes were applied to the SYS transcriptomics dataset. We focused on modules significantly associated with disease, with $p \leq 0.05$ in at least one of the two cohorts. Nine out of 14 modules were associated with disease. Module eigengenes were additionally correlated (Student's asymptotic $p$ value) to dietary factors, and to fecal metabolites. Metabolomics data was log-scaled and cleaned, with zeroes replaced by a fifth of the lowest value per metabolite, and values with over 4 standard deviations from the mean were trimmed to 4 standard deviations from the mean, to avoid extreme values driving correlations. Benjamini–Hochberg FDR correction was applied separately to diet and metabolomics results. Correlations with FDR $\leq 0.25$ were considered significant. This module eigengenes correlation analysis was performed independently for the Sheba and SYS datasets. ToppGene[65] and ToppCluster software were used to perform Gene Set Enrichment Analyses (GSEA) of the protein-coding genes within the modules in the WGCNA TI analysis.

For each of the Sheba TI transcriptomics WGCNA disease-associated modules, we defined modules associated with metabolites as fecal metabolites with significant correlation (FDR $\leq 0.25$) to that module eigengene. An exact binomial test was used to test the consistency of metabolites' correlation to TI WGCNA modules, between all samples and CD samples only in the Sheba cohort. For the 416 significant module-metabolites correlation calculated using all samples, a correlation was considered consistent if it changed in the same direction in CD samples, looking at Spearman's rho. 406 out of 416 correlations were consistent. The probability of success was calculated as $p_{all}*p_{cd} + (1 - p_{all})*(1 - p_{cd})$, with $p_{all}$ representing the percentage of positive correlations in all samples, and $p_{cd}$ representing the percentage of positive correlations in CD samples, to account for the unbalanced positive to negative correlations ratio. HAllA (Hierarchical All-against-All significance testing) version 0.8.20 was used to identify potential correlations between these modules associated metabolites and Israeli FFQ components, stool 16S ASVs, and stool metagenomics (MGX species, pathways, and ECs). HAllA was used with Spearman correlation and FDR cutoff of 0.25.

## Multi-omic analysis

We applied sparse Partial Least Squares (sPLS) regression between pairs of omics[19] including -metagenomics (species, pathways, and ECs), stool metabolomic, and host TI transcriptomics (using disease-related WGCNA modules' PC1 values). This method aims to maximize the shared variation between a pair of omics while accounting for the sparsity in the data. As a measure of the identified shared variation, we calculated the Spearman correlation between the first components of each omic. Significance was evaluated by generating 100 permutations of the feature table, and counting the number of permutations that yielded a higher correlation value than the one calculated for the original data. DIABLO[20] (Data Integration Analysis for Biomarker discovery using Latent variable approaches for Omics studies) was used to simultaneously maximize the shared variation between each omics and differentiate between the health conditions. To uncover the relationship between the omics features and the disease state, we focused on the 10 loadings with the highest correlation to the first and second components. Both sPLS regression and DIABLO were calculated using the MixOmics package[66]. According to the recommendations in the MixOmics package, the number of components was chosen by a minimal balanced error rate using the centroid distance method. In addition, HAllA version 0.8.20 was used to identify potential correlations between China rural and rural-urban metabolites, FFQ components, and stool 16S ASVs, using Spearman correlation and FDR cutoff of 0.25.

## Statistical analysis

Statistics used for transcriptomics, microbiome, and metabolomics were performed in R, and details are under these specific sections. Overall, Pearson's chi-square test or Fisher's exact test was used for categorical variables, Spearman's rank correlation was used for continuous variables, and the Mann–Whitney $U$ test for categorical variables, with Benjamini–Hochberg Procedure for FDR correction using 0.1 or 0.25 as indicated.

## Reporting summary

Further information on research design is available in the Nature Portfolio Reporting Summary linked to this article.

## Data availability

Source datasets are provided. RNASeq Israel and China datasets generated in this study have been deposited in the GEO database under accession code: GSE199906 and GSE233900, respectively. The 16S amplicon sequencing dataset generated in this study has been deposited in BioProject under accession code: PRJNA978342, Microbial shotgun sequencing generated in this study has been deposited BioProject under accession code: PRJNA1056458. Fecal metabolomics datasets: processed data of the predefined library used in this study are provided in Supplementary Dataset 7. Raw datasets generated in this study have been deposited in Metabolomics Workbench[67] under accession code: ST003161 (https://doi.org/10.21228/M8KX6R). Any additional information is available upon request to the corresponding author (yael.haberman@cchmc.org).

## Code availability

The code is available at https://github.com/ShebaMicrobiomeCenter/SOURCE[68].

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

## Acknowledgements

We thank all study participants. Support for the SOURCE study was provided by the Helmsley Charitable Trust (E.E., S.B.H., Y.H.). Other support included the ERC starting grant (Y.H., grant No. 758313), Tel Aviv University's Colton Center for Autoimmune Diseases (E.B., Y.H.), the Israel Science Foundation (Y.H., grant No. 785/22), the I-CORE program (Y.H., grants No. 41/11), Israel Science, Culture, and Sport (Y.H., grant no. 4361), and NIDDK P30 DK078392 (Integrative Morphology and Gene Expression Cores). The funding sources did not play a role in the writing of the manuscript or the decision to submit it for publication, and did not play a role in data collection, analysis, interpretation; trial design; patient recruitment; or any aspect pertaining to the study.

## Author contributions

R.F., E.E., M.C., S.B.H. and Y.H. conceived and designed the study, analyzed the data, received funding, and wrote the first draft of the manuscript. T.B., A.A., N.L., H.S., R.M., R.H., I.T., Y.A., S.Z., G.E., T.C.L. and E.B. generated and analyzed the data and participated in drafting the manuscript. K.A.S., A.M., O.P., M.Y., B.A., T.S.S., L.D., O.K.L. and E.G. collected and analyzed data, recruited patients, and participated in drafting the manuscript. All authors had access to study data and approved the decision to submit the manuscript.

## Competing interests

The following authors have no conflict of interest: T.B., R.F., A.A., N.L., H.S., R.M., R.H., I.T., Y.A., K.A.S., S.Z., G.E., A.M., O.P., M.Y., B.A., T.S.S., L.D., O.K.L., E.G., E.B., M.C., S.B.H. and Y.H. The following authors report conflict of interest: E.E. is a scientific cofounder of DayTwo and BiomX, and an advisor to Hello Inside, Igen, Purposebio and Aposense, an editorial board member of CHM. T.C.L. is an advisor to and receives research contracts from Denali Therapeutics and Interline Therapeutics.
