## [Peer Review File · Nature Communications]

Reviewers' Comments:

Reviewer #1:

Remarks to the Author:

This manuscript describes a comprehensive multi-omic study exploring factors linking rural-urban transitions to Crohns disease. This included the characterization of the diet, microbiota, metabolome and transcriptome of individuals from China and Isreal living in rural and urban areas with and without Crohns. Overall, this is a well presented, thorough and methodologically sound body of work. The findings shed light on the potential factors linking the transition from rural to urban lifestyles to increasing prevalence of CD, highlighting potential targets for future interventions. I believe the work to be of sufficient quality and impact to merit its publication in Nature Communications.

I have the following minor comments regarding the manuscript:

Enteric infection burden is expected to differ between rural and urban sites. Infection status will impact on the intestinal transcriptome, microbiota, and the metabolome. This could be contributing to variation between the rural and urban settings independent of diet. Was this measured?

Figure 1B is not informative and should be removed.

The loadings in Figures 3A and 3B cannot be read and should be increased in size.

Figure 3D, the heatmaps do not provide the reader with a clear result. Difficult to see a difference. Could be moved to the supplementary information.

Line 299: there is no Figure 4H.

The model diagnostics were not reported for the sPLS or the DIABLO models.

IOIBD needs to be defined the first time it is mentioned (line 175).

Reviewer #2:

Remarks to the Author:

Braun, Feng and collaborators study 380 individuals from three different sample sites, including Crohn's disease (CD) patients from two different countries. The different groups present significant differences in environmental exposures and dietary habits, with an interesting itemization of the Chinese rural individuals in a range of rural to urban lifestyles. Through characterization of the fecal omic samples, they show that time spent in urban areas leads to a reduction in microbial health indexes and alterations in the fecal metabolome. Interestingly, they find strong overlap in the direction of changes observed in rural to urban comparisons and the differences observed in patients vs. non Crohn's individuals. This suggests that the fecal alterations observed in CD represent an extreme (higher than average) departure from the microbiota and metabolomic compositions associated with more traditional lifestyles.

They next characterize variability in dietary patterns that are consistent across the subgroups, observing a range of associations between microbial variations and dietary factors that overlap with the changes observed in CD. Analysis of the gut microbiota and metabolome shows the same patterns seen for the fecal samples, highlighting strong individual-specific components. Finally, they characterize the CD ileal transcriptome to find associations between specific dysregulated pathways and particular microbiota, metabolomic and diet components. They find a wealth of associations between CD gene modules and specific dietary factors and metabolite groups. Besides a set of specific hypotheses, they observe an intriguing pattern that suggests diet is associated with epithelial functions whereas microbiota associates with global immune responses.

The study generated a wealth of omic datasets to understand the similarities in gut changes observed in rural to urban transition and Crohn's disease, revealing a wealth of novel and

replicated associations between diet, clinical factors, and gene regulatory components. Although many of the observations are seemingly new, they use publicly available datasets and repositories to strengthen the evidence. The overall setting is very interesting, as it includes (i) analyses of several omics that capture from basic cell activity to the effects of lifestyle, (ii) ability to connect with diet and lifestyle, and (iii) availability of individuals undergoing a transition that cannot be captured in classical studies in Western Europe or North America. The study definitively provides a valuable framework to generate integrative hypotheses on CD with potential for targeted interventions or therapies.

In turn, the cross-sectional and overall correlation-based nature of the study does not permit to infer underlying mechanisms or assign causality or directionality for most of the findings. Moreover, because only CD patients are included, it is not possible to clarify the transferability of the observed patterns to other diseases associated with modern lifestyle. Regardless, the overall balance is clearly positive, and the work here represents an important addition to the community.

I have not found any methodological flaw in the analyses, and the overall analytical approximation looks correct. I could envision further analyses, even more so considering the sparsity of some of the data (eg microbiota) and the large amount of significant associations (eg. between metabolites and microbiota). For instance, it would be interesting to perform further clustering within each group to detect meaningful subclusters of individuals. I suspect that large covariance within and heterogeneity among the subgroups may play an important role that is overlooked in pairwise comparisons. However, it would be unfair to start asking for deeper comparisons when so many novel hypothesis-triggering associations are already reported. The authors use a classical approach based on comparisons among groups that is obviously correct, and it is important to report these observations in a single unified manuscript. The dataset is rich and novel, and making it publicly available opens the venue for future explorations by the community. Although the omic datasets have been made available, I would suggest that the authors find a way to share the full dataset to maximize the potential for future impact. Finally, generation of genetic data for these individuals could be useful to explore potential associations between genetic risk, environmental factors, and the transition to modern lifestyle.

In summary, this dataset represents an exciting addition for the study of Crohn's disease, it provides a path for similar studies in other diseases associated with modern lifestyle, and the analyses of multi-omic data in light of exposure factors uncovers a wealth of novel associations that will be explored in future mechanistic studies. I don't have any suggestion or request for change, and overall would like to congratulate the authors for this interesting piece of work.

Reviewer #3:

Remarks to the Author:

Major comment:

It is a noble aim to apply multi-omics analyses to understand CD, but it is a challenge to aim at investigating rural versus urban living, exposure to soft drinks and other nutrients, exposure to farm animals, siblings, gut microbiome composition, transcriptomics etc. and the interplay between all these variables in only 380 individuals (of which only 65 are CD cases and 315 are controls?) representing two different cohorts from two different countries, hence of different ethnicities, and with further subdivision of at least one of the cohorts according to rural versus urban living (with no cases in the rural arm?). And further adding a second definition of rural versus urban living/exposure in analyses.

This may affect the generalizability of results and it challenges the presentation of data.

Thank you for the thoughtful review of our manuscript entitled "**Diet-omics in the Study of Urban and Rural Crohn disease Evolution (SOURCE) cohort**", and for the opportunity to submit a revised manuscript.

Below is a summary of the response, followed by a detailed point-by-point response to the reviewers' comments.

Overall, the reviewers thought that this is a well-presented, thorough, and methodologically sound body of work and that the work has sufficient quality and impact to merit its publication in Nature Communications (reviewer 1), and that the dataset represents an exciting addition for the study of Crohn's disease, it provides a path for similar studies in other diseases associated with modern lifestyle, and the analyses of multi-omic data in light of exposure factors uncovers a wealth of novel associations that will be explored in future mechanistic studies (reviewer 2). Reviewer 3 referred to the complexity linked with ethnicity, exposure, etc but as indicated by the other reviewers we tried to address these accordingly, still generating exciting and novel hypotheses.

All relevant datasets are available. RNASeq Israel and China datasets were deposited in GEO: GSE199906 and GSE233900, respectively. The 16S amplicon sequencing dataset was deposited in BioProject PRJNA978342. Microbial shotgun sequencing was deposited in BioProject PRJNA1056458. Fecal metabolomics datasets are available as Supplementary Dataset 7. The code is available at <https://github.com/Tzipisb/SOURCE>.

A point-by-point response to the reviewers' comments

Reviewer #1 (Remarks to the Author):

This manuscript describes a comprehensive multi-omic study exploring factors linking rural-urban transitions to Crohn's disease. This included the characterization of the diet, microbiota, metabolome and transcriptome of individuals from China and Israel living in rural and urban areas with and without Crohn's. Overall, this is a well presented, thorough and methodologically sound body of work. The findings shed light on the potential factors linking the transition from rural to urban lifestyles to increasing prevalence of CD, highlighting potential targets for future interventions. I believe the work to be of sufficient quality and impact to merit its publication in Nature Communications.

Response: We thank the reviewer for acknowledging the scope analyses.

I have the following minor comments regarding the manuscript:

Enteric infection burden is expected to differ between rural and urban sites. Infection status will impact on the intestinal transcriptome, microbiota, and the metabolome. This could be contributing to variation between the rural and urban settings independent of diet. Was this measured?

Response: Enteric infection burden was not directly tested in this cohort. This was added as a limitation to the discussion. We have captured CRP, which indirectly indicates potential infection. CRP levels were mostly not different between rural and rural-urban this is indicated in Dataset S1. "Enteric infection burden was not directly tested, but CRP levels that indirectly indicate potential infections were not different between rural and rural-urban (**Dataset S1**)."

Figure 1B is not informative and should be removed.

Response: Figure 1B was removed as suggested.

The loadings in Figures 3A and 3B cannot be read and should be increased in size.

Response: loading font size was increased in Figures 3A and 3B

Figure 3D, the heatmaps do not provide the reader with a clear result. Difficult to see a difference. Could be moved to the supplementary information.

Response: This was moved as suggested to the supplementary information (Figure S3).

Line 299: there is no Figure 4H.

Response: we apologize this was corrected to Figure S4H.

The model diagnostics were not reported for the sPLS or the DIABLO models.

Response: We thank the reviewer for this comment, we added this information in Supplementary Dataset 8 with complete information on the sPLS Spearman correlation p-values and rhos. We further added information on the variance explained by the DIABLO models in the results. “we applied DIABLO²⁰. This analysis demonstrated high shared explained variance between the different omics, ranging from 13% - 53% (metabolomics 24%, metagenomics 13%, transcriptomics 53%, and food frequency questioners 22%) and showed an overall separation of CD from controls based on each omic (Fig. 5g).”

IOIBD needs to be defined the first time it is mentioned (line 175).

Response: IOIBD is now defined the first time it is mentioned.

Reviewer #2 (Remarks to the Author):

Braun, Feng and collaborators study 380 individuals from three different sample sites, including Crohn's disease (CD) patients from two different countries. The different groups present significant differences in environmental exposures and dietary habits, with an interesting itemization of the Chinese rural individuals in a range of rural to urban lifestyles. Through characterization of the fecal omic samples, they show that time spent in urban areas leads to a reduction in microbial health indexes and alterations in the fecal metabolome. Interestingly, they find strong overlap in the direction of changes observed in rural to urban comparisons and the differences observed in patients vs. non Crohn's individuals. This suggests that the fecal alterations observed in CD represent an extreme (higher than average) departure from the microbiota and metabolomic compositions associated with more traditional lifestyles.

They next characterize variability in dietary patterns that are consistent across the subgroups, observing a range of associations between microbial variations and dietary factors that overlap with the changes observed in CD. Analysis of the gut microbiota and metabolome shows the same patterns seen for the fecal samples, highlighting strong individual-specific components. Finally, they characterize the CD ileal transcriptome to find associations between specific dysregulated pathways and particular microbiota, metabolomic and diet components. They find a wealth of associations between CD gene modules and specific dietary factors and metabolite groups. Besides a set of specific hypotheses, they observe an intriguing pattern that suggests diet is associated with epithelial functions

whereas microbiota associates with global immune responses.

The study generated a wealth of omic datasets to understand the similarities in gut changes observed in rural to urban transition and Crohn's disease, revealing a wealth of novel and replicated associations between diet, clinical factors, and gene regulatory components. Although many of the observations are seemingly new, they use publicly available datasets and repositories to strengthen the evidence. The overall setting is very interesting, as it includes (i) analyses of several omics that capture from basic cell activity to the effects of lifestyle, (ii) ability to connect with diet and lifestyle, and (iii) availability of individuals undergoing a transition that cannot be captured in classical studies in Western Europe or North America. The study definitively provides a valuable framework to generate integrative hypotheses on CD with potential for targeted interventions or therapies.

In turn, the cross-sectional and overall correlation-based nature of the study does not permit to infer underlying mechanisms or assign causality or directionality for most of the findings. Moreover, because only CD patients are included, it is not possible to clarify the transferability of the observed patterns to other diseases associated with modern lifestyle. Regardless, the overall balance is clearly positive, and the work here represents an important addition to the community.

I have not found any methodological flaw in the analyses, and the overall analytical approximation looks correct. I could envision further analyses, even more so considering the sparsity of some of the data (eg microbiota) and the large amount of significant associations (eg. between metabolites and microbiota). For instance, it would be interesting to perform further clustering within each group to detect meaningful subclusters of individuals. I suspect that large covariance within and heterogeneity among the subgroups may play an important role that is overlooked in pairwise comparisons. However, it would be unfair to start asking for deeper comparisons when so many novel hypothesis-triggering associations are already reported. The authors use a classical approach based on comparisons among groups that is obviously correct, and it is important to report these observations in a single unified manuscript. The dataset is rich and novel, and making it publicly available opens the venue for future explorations by the community. Although the omic datasets have been made available, I would suggest that the authors find a way to share the full dataset to maximize the potential for future impact. Finally, generation of genetic data for these individuals could be useful to explore potential associations between genetic risk, environmental factors, and the transition to modern lifestyle.

In summary, this dataset represents an exciting addition for the study of Crohn's disease, it provides a path for similar studies in other diseases associated with modern lifestyle, and the analyses of multi-omic data in light of exposure factors uncovers a wealth of novel associations that will be explored in future mechanistic studies. I don't have any suggestion or request for change, and overall would like to congratulate the authors for this interesting piece of work.

Response: We thank the reviewer for this thorough review.

All relevant datasets were made available. RNASeq Israel and China datasets were deposited in GEO: GSE199906 and GSE233900, respectively. The 16S amplicon sequencing dataset was deposited in BioProject PRJNA978342. Microbial shotgun sequencing was deposited in BioProject PRJNA1056458. Fecal metabolomics datasets are available as dataset S7.

The code is available at <https://github.com/Tzipisb/SOURCE>

We attempted to perform further clustering using the microbiota and/or metabolites and to perform enrichment for the clusters with the metadata within each group, and we were able to validate the signals linked with time spent in urban environments, fat, protein, and iron consumption, which further validate

our reported results. However, this did not elude to newer subclusters. As the reviewer already summarized, many novel hypothesis-triggering associations have already been reported. Further validation and mechanistical testing of some of these associations will be part of future work.

Reviewer #3 (Remarks to the Author):

Major comment:

It is a noble aim to apply multi-omics analyses to understand CD, but it is a challenge to aim at investigating rural versus urban living, exposure to soft drinks and other nutrients, exposure to farm animals, siblings, gut microbiome composition, transcriptomics etc. and the interplay between all these variables in only 380 individuals (of which only 65 are CD cases and 315 are controls?) representing two different cohorts from two different countries, hence of different ethnicities, and with further subdivision of at least one of the cohorts according to rural versus urban living (with no cases in the rural arm?). And further adding a second definition of rural versus urban living/exposure in analyses. This may affect the generalizability of results and it challenges the presentation of data.

Response: We thank the reviewer for noting the goal of the study and for acknowledging the challenges linked with the cohort size, the ethnicities studied and the subdivision of the subcohorts. However, we were able to identify significant observations, and some are highlighted here:

- 1) Rural-to-urban transition mirror changes seen in CD. By stratifying all rural residents (n=162) by the time they spent in urban environments we highlighted changes in the rural-urban microbiome and metabolites that were also observed in CD compared to healthy controls. We acknowledge that rural-urban transition and its related changes in the microbiome and metabolites are not directly causing CD, as this entity is a multifactorial and complex disease. However, the fact that CD is linked with urbanization and our finding that these changes in the metabolites and gut microbiome seen with rural-urban transition highlight their potential contribution to CD.
- 2) In addition to the rural-to-urban transition, other exposures and diet components were significantly linked with microbial variations independently within the different groups. Specifically, we show that within controls in China, fat and iron were among the more dominant factors. Surprisingly, taxa that were decreased with increasing iron or fat consumption in controls were enriched for taxa previously shown to be decreased in CD, and those taxa that were higher with increasing fat and iron were enriched for bacteria usually seen in saliva samples, also previously linked with CD.
- 3) By applying an unbiased approach to the mucosal ileal transcriptomics data, we defined different gene co-expression modules with assigned enriched functional pathways that separate the disease signal into more distinct functional modules. This highlighted host metabolic and immune gene expression modules on the health-disease axis that were linked to potential unexpected protective dietary exposures (coffee, manganese, vitamin D), fecal metabolites, and the microbiome. We show that bacteria-associated metabolites were primarily linked with host immune modules, whereas diet-linked metabolites were associated with host epithelial metabolic functions.

We already acknowledge some of the limitations suggested by the reviewer in the discussion “Limitations of our work include the more limited metabolomics and transcriptomics data, which was only available for a subset of cohorts, and the use FFQ, which is frequently used to monitor dietary habits but has limitations. Since CD is rare in rural areas, our cohorts did not include analyses from this subgroup and CD patients were from urban areas. We used throughout FDR cutoff of 0.25, but we supplemented the results when possible, with independently validated, we indicated the features that had FDR cutoff of 0.1, and p and FDR values are given in the suppl. datasets. Additionally, due to limitations associated with the COVID-19 outbreak, samples from China and Israel were processed

locally within each country of origin, but both cohorts were analyzed using a similar pipeline and as in the case of the metabolomics association with the transcriptomics modules were used as an independent validation.”

We now added “It is possible that with an enlarged cohort in the indicated ethnicities and subcohorts, we could have identified other weaker signals that we were not able to detect with the current cohort size. However, using the current cohort, we were able to find important and substantial results using a rigorous statistical analysis. “

Reviewers' Comments:

Reviewer #1:

Remarks to the Author:

I am satisfied that the authors have addressed my earlier comments.

Reviewer #2:

Remarks to the Author:

I don't have any major comment regarding the response to reviewers. I am happy that they had a look into the subcluster suggestion, and I agree with the authors in that it is not worth including them here. This is a nice piece of work, and the authors should feel proud of it.

Reviewer #3:

Remarks to the Author:

No further comments

We thank the reviewers and editor for the thoughtful review of our manuscript entitled "**Diet-omics in the Study of Urban and Rural Crohn disease Evolution (SOURCE) cohort**".

Overall, the reviewers had no comments after the first round of revision.

Based on the editor's letter, it was noted that Reviewer #3 confidentially pointed out that the reporting in the manuscript needs additional clarity on how many patients and controls were included in all the individual analyses (i.e. adding the sample size information for each experiment) and a thorough discussion of the limitations associated to performing multi-omics analyses on a very small cohort covering two geographically very different sites.

Response: These points were added throughout the results, methods, and figure legends. Limitations are noted in the discussion.

“Limitations of our work include performing multi-omics analyses on a relatively small cohort covering two geographically very different sites and the more limited metabolomics and transcriptomics data, which was only available for a subset of cohorts, and the use FFQ, which is frequently used to monitor dietary habits but has limitations. Since CD is rare in rural areas, our cohorts did not include analyses from this subgroup and CD patients were from urban areas. We used throughout FDR cutoff of 0.25, but we supplemented the results when possible, with independently validated, we indicated the features that had FDR cutoff of 0.1, and p and FDR values are given in the suppl. datasets. It is possible that with an enlarged cohort in the indicated ethnicities, we could have identified other weaker signals that we were not able to detect with the current cohort size. However, using the current cohort, we were able to find important and substantial results using rigorous analyses. We did not include single-cell transcriptomics, but there are advantages to using whole mucosal gut biopsies, which are used for diagnosis and follow-up in the clinical setting. Enteric infection burden was not directly tested, but CRP levels that indirectly indicate potential infections were not different between rural and rural-urban (**Supplementary Dataset1**). Additionally, due to limitations associated with the COVID-19 outbreak, samples from China and Israel were processed locally within each country of origin, but both cohorts were analyzed using a similar pipeline and as in the case of the metabolomics association with the transcriptomics modules were used as an independent validation.”

REVIEWERS' COMMENTS

Reviewer #1 (Remarks to the Author):

I am satisfied that the authors have addressed my earlier comments.

Reviewer #2 (Remarks to the Author):

I don't have any major comment regarding the response to reviewers. I am happy that they had a look into the subcluster suggestion, and I agree with the authors in that it is not worth including them here. This is a nice piece of work, and the authors should feel proud of it.

Reviewer #3 (Remarks to the Author):

No further comments

Response: Thank you for the thoughtful review of our manuscript.